# Approximate Kernel Density Estimation under Metric-based Local Differential Privacy

**Yi Zhou**[1]  **Yanhao Wang**[*1]  **Long Teng**[1]  **Qiang Huang**[2]  **Cen Chen**[1]

[1]School of Data Science and Engineering, East China Normal University, Shanghai, China
[2]School of Computing, National University of Singapore, Singapore

## Abstract

Kernel Density Estimation (KDE) is a fundamental problem with broad machine learning applications. In this paper, we investigate the KDE problem under Local Differential Privacy (LDP), a setting in which users privatize data on their own devices before sending them to an untrusted server for analytics. To strike a balance between ensuring local privacy and preserving high-utility KDE results, we adopt a relaxed definition of LDP based on metrics (mLDP), which is suitable when data points are represented in a metric space and can be more distinguishable as their distances increase. To the best of our knowledge, approximate KDE under mLDP has not been explored in the existing literature. We propose the MLDP-KDE framework, which augments a locality-sensitive hashing-based sketch method to provide mLDP and answer any KDE query unbiasedly within an additive error with high probability in sublinear time and space. Extensive experimental results demonstrate that the MLDP-KDE framework outperforms several existing KDE methods under LDP and mLDP by achieving significantly better trade-offs between privacy and utility, with particularly remarkable advantages on large, high-dimensional data.

## 1 INTRODUCTION

In today's digital era, vast amounts of user-generated data are collected daily by service providers for analysis. Such data are often of a sensitive nature because analytical results on them, even in the form of aggregated statistics or predictions, can leak substantial information about individual users [Dwork and Roth, 2014, Hu et al., 2022]. Therefore, while *effectiveness* and *efficiency* remain fundamental for

---
*Corresponding author.

data analysis tasks, *privacy* has also emerged as a crucial concern.

In the realm of privacy-preserving data analysis, local differential privacy (LDP) [Duchi et al., 2013] offers a theoretically rigorous definition of privacy that allows an untrusted server to collect and analyze user data privately with guarantees. In particular, LDP is the local model of differential privacy (DP) [Dwork et al., 2006], the de facto standard of privacy preservation, wherein users perturb data on their own devices before sending them to the server. Informally, a mechanism is locally differentially private if its output is indistinguishable for any given pair of input values. This guarantees a level of protection against information leakage in any individual record measured by the privacy parameter $\varepsilon > 0$. A smaller value of $\varepsilon$ implies a lower level of distinguishability and more reliable privacy protection. Despite providing provably strong privacy guarantees, LDP is meanwhile known to incur considerable losses in the quality of analytical results [Cormode et al., 2018, Xiang et al., 2020]. Consequently, it has become a prominent challenge to accurately analyze large-scale data sets while maintaining local privacy [Cormode et al., 2018, Duchi et al., 2013, Kairouz et al., 2016, Wang et al., 2019, Li et al., 2020].

In this paper, we study the kernel density estimation (KDE) problem [Parzen, 1962], which is a cornerstone of numerous machine learning applications, including clustering [Hinneburg and Keim, 2003], anomaly detection [Hu et al., 2020], and visualization [Chan et al., 2021, 2022], in a local privacy setting. KDE is an unsupervised technique for estimating a probability density function from data points, offering insights into the underlying data distribution and patterns. Given a data set $\mathcal{D} \subseteq \mathbb{R}^m$ of $n$ points in an $m$-dimensional space and a kernel function $k : \mathbb{R}^m \times \mathbb{R}^m \mapsto [0,1]$, the kernel density at a query point $\boldsymbol{q} \in \mathbb{R}^m$ is defined as

$$\text{KDE}_{\mathcal{D}}(\boldsymbol{q}) = \frac{1}{n} \sum_{\boldsymbol{x} \in \mathcal{D}} k(\boldsymbol{x}, \boldsymbol{q}).$$

Exact computation of $\text{KDE}_{\mathcal{D}}(\boldsymbol{q})$, requiring $O(nm)$ time and space, is costly for large, high-dimensional datasets.

Recent advances have focused on developing methods for approximate KDE that are more efficient in terms of time and memory [Muandet et al., 2017, Charikar and Siminelakis, 2017, Siminelakis et al., 2019, Phillips and Tai, 2018, Backurs et al., 2019, Coleman and Shrivastava, 2020, Lei et al., 2021]. Particularly noteworthy is the exploration of the intrinsic connection between KDE and locality-sensitive hashing (LSH) [Charikar and Siminelakis, 2017, Siminelakis et al., 2019, Backurs et al., 2019]. This connection has led to several sketch-based methods [Coleman and Shrivastava, 2020, Lei et al., 2021] to estimate the density of LSH kernels within sublinear time and memory space while also providing approximation guarantees. Moreover, the *mergeability* of the sketches significantly improves the applicability of sketch-based methods, especially in distributed and streaming models [Coleman and Shrivastava, 2020, Lei et al., 2021]. This enables the construction of a sketch for the entire data set $\mathcal{D}$ by seamlessly combining the sketches created from its disjoint subsets. Although sketch-based KDE methods do not require storing original data, they do not inherently guarantee differential privacy. This is because an adversarial server can potentially recover users' data from the hash values they send [Fernandes et al., 2021]. In addition, existing differentially private sketches for KDE [Coleman and Shrivastava, 2021, Wagner et al., 2023] are tailored for the centralized setting, where user data should first be gathered by a *trusted* server, and then a sketch is constructed with perturbation from the original data to answer KDE queries privately. However, these methods are not adaptable to the LDP setting, where user data must be privatized prior to collection.

## 1.1 MAIN CONTRIBUTIONS

To fill this gap, we first attempt to approximate KDE subject to LDP constraints. We observe that the conventional LDP notion, which demands indistinguishability between any pair of inputs, is too stringent for the KDE problem. It often leads to prohibitive errors in the KDE query results, even when a substantial privacy budget is allocated. To strike a balance between maintaining an adequate level of local privacy and preserving high-utility KDE results, we opt for a more relaxed metric-based variant of LDP, known as *local $d_\chi$-privacy* [Chatzikokolakis et al., 2013, Alvim et al., 2018]. Specifically, metric-based LDP (mLDP) quantifies the distinguishability between any two data points $x, x'$ in relation to their distance $d_\chi(x, x')$ within a given metric space $\chi$. A point $x$ becomes more distinguishable from another point $x'$ as $d_\chi(x, x')$ increases and vice versa. In this way, mLDP allows the server to collect approximate information from users while protecting the exact values of individual data points. This characteristic of mLDP is particularly compatible with the nature of KDE because many kernel classes, including common LSH kernels, are defined directly on metric distances. For example, the $l_1$-

and $l_2$-LSH kernels are derived from the Manhattan and Euclidean distances [Coleman and Shrivastava, 2020], while the angular kernel corresponds to the angular distance [Lei et al., 2021]. Furthermore, mLDP has a distinct advantage over traditional LDP in preserving data distribution, which is essential for high-utility KDE results.

To provide mLDP, we design a general framework that augments sketch-based KDE methods by introducing the generalized randomized response (GRR) mechanism [Kairouz et al., 2016] for users to perturb hash values before sending them to the server. Building on this framework, we introduce an unbiased estimator that enables the server to accurately answer KDE queries using the sketch built by aggregating the perturbed hash values from users. Our theoretical analysis shows that the user-level mechanism for computing the hash values provides mLDP with high probability. Moreover, any KDE result provided by the server, calculated in sublinear time and space, has bounded additive errors, also with high probability. To the best of our knowledge, this is the first method for approximate KDE under mLDP. The main contributions are summarized as follows.

- We formally define the problem of approximate kernel density estimation (KDE) under metric-based local differential privacy (mLDP). (Section 2)

- We propose a novel MLDP-KDE framework and analyze its privacy guarantee, approximation bound, and complexity theoretically. (Section 3)

- We conduct extensive experiments on five real-world and synthetic data sets to evaluate the performance of the MLDP-KDE framework. The results confirm its superiority over existing methods for KDE under LDP and mLDP by achieving significantly better privacy-utility trade-offs and demonstrating better scalability on large, high-dimensional data sets. (Section 4)

## 1.2 RELATED WORK

**Approximate KDE on High-Dimensional Data.** The KDE [Parzen, 1962] problem requires expensive $O(nm)$ time and space for exact computation. To mitigate this issue, sublinear methods have emerged for approximating KDE on large, high-dimensional data, which can be roughly categorized into sampling- and sketch-based methods.

Sampling-based methods approximate the KDE over randomly sampled subsets to compute the KDE on the entire dataset. Although Muandet et al. [2017] and subsequent work [Charikar and Siminelakis, 2017, Cortes and Scott, 2017, Siminelakis et al., 2019, Phillips and Tai, 2018, Backurs et al., 2019] have explored efficient sampling methods, they are not suitable for the local model, as they require the full data set for computation.

Sketch-based methods leverage LSH schemes to build a succinct array of counters for a data set, and they approx-

imate the KDE by computing the hash values for a query point and aggregating the corresponding counters. Coleman and Shrivastava [2020] proposed Repeated Array-of-Counts Estimator (RACE), a sketch-based method for KDE. Subsequently, Lei et al. [2021] devised a more efficient KDE sketch for the angular kernel. Due to their manageability, they can be adapted to the local model. However, to the best of our knowledge, they cannot provide any LDP guarantee.

Differentially private KDE (DP-KDE) has also gained much attention. Some function release mechanisms [Hall et al., 2013, Aldà and Rubinstein, 2017] can be adapted for DP-KDE by regarding the kernel function as a generalized linear function. However, they exhibit exponential time complexity w.r.t. the dimensionality $m$, rendering them impractical for high-dimensional data. Efforts have been made to extend sampling- and sketch-based KDE methods to satisfy DP [Coleman and Shrivastava, 2021, Wagner et al., 2023], but they remain limited to a centralized DP setting and cannot be extended to provide local privacy.

**LSH under (Metric-based) LDP.** Locally differentially private LSH schemes also made some progress recently. Aumüller et al. [2020] and Fernandes et al. [2021] independently extended the LSH schemes for Jaccard [Broder et al., 2000] and angular [Charikar, 2002] distances to satisfy metric-based LDP similar to that used in this work. Hu et al. [2023] introduced an LDP algorithm using LSH for federated recommender systems. Our method differs from them in the following four aspects: (1) we target the KDE problem, whereas the aforementioned methods focus on similarity search problems; (2) we address the KDE for Euclidean and more general metric distances beyond Jaccard and angular distances; (3) our mLDP definition, which will be given in Section 2, is different from the privacy concept used in [Aumüller et al., 2020, Hu et al., 2023]; (4) our method offers theoretical utility guarantees, but the methods in [Fernandes et al., 2021, Hu et al., 2023] do not.

**Data Analytics under Metric-based LDP.** The concept of mLDP, initially proposed to protect location privacy and termed as *local $d_\chi$-privacy* and *geo-indistinguishability* [Andrés et al., 2013, Bordenabe et al., 2014, Zhao et al., 2023], has broadened its scope. Though first applied to two-dimensional Euclidean space, it has been expanded to provide privacy guarantees in higher dimensions or other metric spaces. Gu et al. [2019] and Xiang et al. [2020] studied the problem of range counting in multidimensional databases under mLDP. Moreover, mLDP has been adopted for private analyses of various types of unstructured data, including texts [Feyisetan et al., 2020, Yue et al., 2021, Carvalho et al., 2023, Du et al., 2023], images [Fan, 2019], and audio [Han et al., 2020]. More recently, Yang et al. [2022] studied $k$-means clustering under mLDP. Nevertheless, these methods are not directly comparable to ours, and we do not notice any prior work on approximate KDE under mLDP.

## 2  PRELIMINARIES

This section presents the background of kernel density estimation (KDE), locality-sensitive hashing (LSH) kernels, and metric-based local differential privacy (mLDP), and formally defines the problem studied in this work.

**Kernel Density Estimation.** We denote $\mathcal{D}$ as a data set of $n$ data points in an $m$-dimensional Euclidean space $\mathbb{R}^m$. A *kernel* is defined as a function $k : \mathbb{R}^m \times \mathbb{R}^m \mapsto [0, 1]$ that quantifies the similarity of two points in $\mathbb{R}^m$. For any given query point $\boldsymbol{q} \in \mathbb{R}^m$, the *kernel density estimation* (KDE) for a data set $\mathcal{D}$, represented as $\mathrm{KDE}_{\mathcal{D}} : \mathbb{R}^m \mapsto [0, 1]$, is defined by $\mathrm{KDE}_{\mathcal{D}}(\boldsymbol{q}) = \frac{1}{n} \sum_{\boldsymbol{x} \in \mathcal{D}} k(\boldsymbol{x}, \boldsymbol{q})$. Our goal is to approximate $\mathrm{KDE}_{\mathcal{D}}(\boldsymbol{q})$ for any $\boldsymbol{q} \in \mathbb{R}^m$. We aim to achieve this through a randomized $(\alpha, \eta)$-approximation as below.

**Definition 1** (($\alpha, \eta$)-Approximate KDE). *Let $\alpha, \eta \in (0, 1)$. Given a data set $\mathcal{D} \subset \mathbb{R}^m$, a query point $\boldsymbol{q} \in \mathbb{R}^m$, and a kernel function $k(\cdot, \cdot)$, $\widehat{\mathrm{KDE}}_{\mathcal{D}}(\boldsymbol{q})$ is an $(\alpha, \eta)$-approximation of $\mathrm{KDE}_{\mathcal{D}}(\boldsymbol{q})$ if $\Pr[|\widehat{\mathrm{KDE}}_{\mathcal{D}}(\boldsymbol{q}) - \mathrm{KDE}_{\mathcal{D}}(\boldsymbol{q})| \le \alpha] \ge 1 - \eta$.*

Moreover, the approximation bound to $\mathrm{KDE}_{\mathcal{D}}(\boldsymbol{q})$, as outlined in Definition 1, should be achieved with space and query time complexities that are sublinear w.r.t. $n$ and polynomial w.r.t. $m$. This requirement is critical for efficient processing of large, high-dimensional data sets.

**Locality-Sensitive Hashing Kernel.** Let $d : \mathbb{R}^m \times \mathbb{R}^m \mapsto \mathbb{R}_{\ge 0}$ be a distance function to measure the *dissimilarity* between any two points in $\mathbb{R}^m$. We call $d(\cdot, \cdot)$ a metric if it satisfies the axioms of non-negativity, identity of indiscernibles, symmetry, and triangle inequality. An LSH family $\mathcal{H}$ w.r.t. a metric distance $d(\cdot, \cdot)$ is a family of hash functions $h : \mathbb{R}^m \mapsto \mathbb{Z}$ such that for any two points $\boldsymbol{x}, \boldsymbol{x}' \in \mathbb{R}^m$, the probability that $h(\boldsymbol{x}) = h(\boldsymbol{x}')$ monotonically decreases with $d(\boldsymbol{x}, \boldsymbol{x}')$. Formally, $\Pr_{h \in \mathcal{H}}[h(\boldsymbol{x}) = h(\boldsymbol{x}')] = f(d(\boldsymbol{x}, \boldsymbol{x}'))$, where $f(\cdot)$ is a monotonically decreasing collision probability function. In many LSH families, this collision probability function forms a positive semidefinite radial kernel [Coleman and Shrivastava, 2020], which is referred to as an *LSH kernel*, i.e., $k(\boldsymbol{x}, \boldsymbol{x}') = f(d(\boldsymbol{x}, \boldsymbol{x}'))$. Notable examples of LSH schemes, including the Signed Random Projection (SRP) LSH for the angular distance [Charikar, 2002] and the 1-stable or 2-stable LSH scheme for the Manhattan ($l_1$-) or Euclidean ($l_2$-) distance [Datar et al., 2004, Huang et al., 2017], all induce useful LSH kernels.

In this paper, we focus on the $l_2$-LSH kernel for the Euclidean distance.[1] Specifically, a hash function in the 2-stable LSH family [Datar et al., 2004] is defined as $h(\boldsymbol{x}) = \lfloor \frac{\boldsymbol{a} \cdot \boldsymbol{x} + b}{\omega} \rfloor$, where $\boldsymbol{a}$ is a vector drawn from the standard $m$-dimensional Gaussian distribution, $b$ is a scalar drawn from the uniform distribution $\mathcal{U}(0, \omega)$, and $\omega > 0$ is the bandwidth. Let $d(\boldsymbol{x}, \boldsymbol{x}') = \|\boldsymbol{x} - \boldsymbol{x}'\|_2$. The $l_2$-LSH kernel is

---

[1]Note that our results can be extended to other LSH kernels, as shown in Appendix D.

denoted by a complex function as

$$k(\boldsymbol{x}, \boldsymbol{x}') = 1 - 2\Psi\left(\frac{-\omega}{d(\boldsymbol{x}, \boldsymbol{x}')}\right)$$
$$- \frac{2d(\boldsymbol{x}, \boldsymbol{x}')}{\sqrt{2\pi}\omega}\left(1 - \exp\left(\frac{-\omega^2}{2d^2(\boldsymbol{x}, \boldsymbol{x}')}\right)\right), \quad (1)$$

where $\Psi(\cdot)$ is the cumulative distribution function (CDF) of the standard Gaussian distribution $\mathcal{N}(0, 1)$.

**Metric-based Local Differential Privacy.** Let $\mathcal{V}$ and $\mathcal{Y}$ be the input and output domains of a randomized mechanism $\mathcal{M}$. Given a parameter $\varepsilon > 0$, the mechanism $\mathcal{M} : \mathcal{V} \mapsto \mathcal{Y}$ satisfies $\varepsilon$-local differential privacy ($\varepsilon$-LDP) if for every pair of inputs $v, v' \in \mathcal{V}$ and measurable output $Y \subset \mathcal{Y}$, $\Pr[\mathcal{M}(v) \in Y] \leq e^\varepsilon \cdot \Pr[\mathcal{M}(v') \in Y]$. This ensures that $\mathcal{M}$ cannot reliably distinguish between $v$ and $v'$ for any observed output $Y$, preventing an adversary from inferring the original data. A lower value of $\varepsilon$ indicates a stronger level of privacy, as the probability that the output can be used to infer its corresponding input is reduced.

The GRR [Kairouz et al., 2016] mechanism satisfies $\varepsilon$-LDP for any finite and discrete domain. To facilitate understanding, let $\mathcal{V}, \mathcal{Y}$ represent the input and output domains of the GRR mechanism, both length-$R$ arrays indexed by $[1, 2, \cdots, R]$. For any input value $v \in \mathcal{V}$, the output $\mathcal{M}_{\mathrm{GRR}}(v) \in \mathcal{Y}$ is a random variable sampled as follows:

$$\Pr[\mathcal{M}_{\mathrm{GRR}}(v) = i] = \begin{cases} \frac{e^\varepsilon}{e^\varepsilon + R - 1} & \text{if } i = v, \\ \frac{1}{e^\varepsilon + R - 1} & \text{otherwise.} \end{cases} \quad (2)$$

We consider a metric-based variant of LDP (mLDP), also known as local $d_\chi$-privacy [Chatzikolakis et al., 2013, Alvim et al., 2018], which relaxes the privacy requirements by allowing two data points to become more distinguishable as their distance increases. Formally, a randomized mechanism $\mathcal{M} : \mathcal{V} \mapsto \mathcal{Y}$ satisfies local $d_\chi$-privacy if for any pair of inputs $v, v'$ and any measurable output $Y \subset \mathcal{Y}$,

$$\Pr[\mathcal{M}(v) \in Y] \leq e^{d_\chi(v, v')} \cdot \Pr[\mathcal{M}(v') \in Y],$$

where $d_\chi(\cdot, \cdot)$ is a distance function in the metric space $\chi$.

We aim to devise a randomized mechanism that provides mLDP for KDE using LSH kernels, which takes a data point in $\mathbb{R}^m$ as input and an integer array as output. As an LSH scheme only approximately preserves the original distance, we introduce a new metric distance, $d_\chi(\cdot, \cdot)$, into the mLDP definition by allowing extra errors w.r.t. the original $d(\cdot, \cdot)$, i.e., $d_\chi(\boldsymbol{x}, \boldsymbol{x}') = \mu \cdot d(\boldsymbol{x}, \boldsymbol{x}') + \delta$, where $\mu, \delta > 0$. It is evident that $d_\chi(\cdot, \cdot)$ is a metric as long as $d(\cdot, \cdot)$ is a metric. Furthermore, given the probabilistic nature of LSH schemes, our privacy guarantee is inherently satisfied with a certain probability when LSH functions are chosen randomly, which aligns with the concept of probabilistic differential privacy [Machanavajjhala et al., 2008]. Formally,

**Definition 2** (($d_\chi, \eta$)-mLDP). *For any $\mu, \delta > 0$ and $\eta \in (0, 1)$, a randomized mechanism $\mathcal{M} : \mathbb{R}^m \mapsto \mathbb{Z}^L$ provides $(d_\chi, \eta)$-mLDP iff any two inputs $\boldsymbol{x}, \boldsymbol{x}' \in \mathbb{R}^m$ and output $\boldsymbol{y} \in \mathbb{Z}^L$, $\Pr\left[\frac{\Pr[\mathcal{M}(\boldsymbol{x}) = \boldsymbol{y}]}{\Pr[\mathcal{M}(\boldsymbol{x}') = \boldsymbol{y}]} \leq \exp(d_\chi(\boldsymbol{x}, \boldsymbol{x}'))\right] \geq 1 - \eta$.*

On the basis of all the concepts above, we formally define the problem studied in this paper.

**Definition 3** (Approximate KDE under mLDP). *For a data set $\mathcal{D} \subset \mathbb{R}^m$ and the $l_2$-LSH kernel $k(\cdot, \cdot)$ with a bandwidth $\omega > 0$, build a sketch $\mathcal{S}_\mathcal{D}$ to offer an $(\alpha, \eta)$-approximate $\widehat{\mathrm{KDE}}_\mathcal{D}(\boldsymbol{q})$ for any $\boldsymbol{q} \in \mathbb{R}^m$ under $(d_\chi, \eta)$-mLDP.[2]*

# 3 OUR ALGORITHM

This section introduces MLDP-KDE, an LSH-based framework for KDE under mLDP, and analyzes it theoretically.

## 3.1 THE MLDP-KDE FRAMEWORK

**Overview.** Coleman and Shrivastava [2020] and Lei et al. [2021] have developed LSH-based sketches for approximate density estimation on LSH kernels. They are adaptable to a local computing model, where users independently calculate hash values and send them to a central server, which then aggregates all of them into a comprehensive sketch for KDE. However, this approach does not inherently provide local differential privacy, as the server can potentially infer individual user data from the transmitted hash values.

To address this and align LSH-based sketches with mLDP in Definition 2, we show that it suffices to perturb hash values using GRR [Kairouz et al., 2016] with a specific privacy parameter on the user side before sending them to the server. Remarkably, the sketch composed of perturbed hash values can provide an unbiased KDE at any query point within a bounded additive error with high probability.

**Sketch Construction.** Algorithm 1 depicts the MLDP-KDE sketch construction procedure. Initially, the server randomly selects $L$ LSH functions from the 2-stable LSH scheme [Datar et al., 2004] and sends the hash parameters to users. Then a user with the data point $\boldsymbol{x} \in \mathcal{D}$ generates $L$ integers by hashing $\boldsymbol{x}$ with $L$ LSH functions, which are then rehashed into the range of $[1, R]$ using a scheme consistent across the server and all users. The user independently runs the GRR mechanism on each hash value using the same parameter $\gamma$ determined by the input privacy budget $\varepsilon$, radius $r$, confidence parameter $\eta$, and height and width $L, R$ of the sketch (see Theorem 1 and Corollary 1 in Section 3.2 for the determination of $\gamma$). After obtaining the perturbed hash values $\widehat{H}(\boldsymbol{x})$, the user sends them back to the server. This procedure is called the LSH+GRR mechanism since

---

[2]The two $\eta$'s in the privacy and approximation bounds can take different values but are kept the same in our analysis for simplicity.

**Algorithm 1:** MLDP-KDE Sketch Construction

**Input:** Data set $\mathcal{D}$, bandwidth $\omega$, privacy budget $\varepsilon$, radius $r$, confidence parameter $\eta$, sketch height $L$ and width $R$

**Output:** Sketch $\mathcal{S}_{\mathcal{D}}$

▷ Server side
**for** $i = 1$ **to** $L$ **do**
  Draw a vector of $m$ random variables from $\mathcal{N}(0, 1)$ as $\boldsymbol{a}_i$ and a random variable from $\mathcal{U}(0, \omega)$ as $b_i$;

Send the LSH parameters $\boldsymbol{A} = [\boldsymbol{a}_1, \cdots, \boldsymbol{a}_L]$ and $\boldsymbol{B} = [b_1, \cdots, b_L]$ to each user;

▷ User side with a data point $\boldsymbol{x} \in \mathcal{D}$ on receiving $\boldsymbol{A}$ and $\boldsymbol{B}$

Set $\gamma \leftarrow \varepsilon / \left( \frac{0.8 r L (R-1)}{\omega R} + \sqrt{\frac{L \ln(1/\eta)}{2}} \right)$ or by Corollary 1;
**for** $i = 1$ **to** $L$ **do**
  $h_i(\boldsymbol{x}) \leftarrow \text{Rehash}\left( \lfloor \frac{\boldsymbol{a}_i \cdot \boldsymbol{x} + b_i}{\omega} \rfloor, R \right)$;
  $\widehat{h}_i(\boldsymbol{x}) \leftarrow \mathcal{M}_{\text{GRR}}(h_i(\boldsymbol{x}))$ with parameter $\gamma$;

Report $\widehat{H}(\boldsymbol{x}) = [\widehat{h}_1(\boldsymbol{x}), \cdots, \widehat{h}_L(\boldsymbol{x})]$ to the server;

▷ Server side
Initialize sketch $\mathcal{S}_{\mathcal{D}} \leftarrow \boldsymbol{0}^{L \times R}$;
**foreach** $\boldsymbol{x} \in \mathcal{D}$ **do**
  **for** $i = 1$ **to** $L$ **do**
    $\mathcal{S}_{\mathcal{D}}[i, \widehat{h}_i(\boldsymbol{x})] \leftarrow \mathcal{S}_{\mathcal{D}}[i, \widehat{h}_i(\boldsymbol{x})] + 1$;

**return** $\mathcal{S}_{\mathcal{D}}$;

---

**Algorithm 2:** MLDP-KDE Query Processing

**Input:** Sketch $\mathcal{S}_{\mathcal{D}}$, query point $\boldsymbol{q}$, the same hash and privacy parameters as Algorithm 1, group parameter $L'$

**Output:** Approximation $\widehat{\text{KDE}}_{\mathcal{D}}(\boldsymbol{q})$ of $\text{KDE}_{\mathcal{D}}(\boldsymbol{q})$

**for** $i = 1$ **to** $L$ **do**
  $h_i(\boldsymbol{q}) \leftarrow \text{Rehash}\left( \lfloor \frac{\boldsymbol{a}_i \cdot \boldsymbol{q} + b_i}{\omega} \rfloor, R \right)$;
$\widehat{K} \leftarrow \boldsymbol{0}^{L'}$;
**for** $l = 1$ **to** $L'$ **do**
  **for** $j = 1$ **to** $\frac{L}{L'}$ **do**
    $i \leftarrow (l-1)L' + j$;
    $\widehat{\mathcal{S}}_{\mathcal{D}}[i, h_i(\boldsymbol{q})] \leftarrow \frac{e^{\gamma} + R - 1}{(e^{\gamma} - 1)(R - 1)} \cdot$
    $(\mathcal{S}_{\mathcal{D}}[i, h_i(\boldsymbol{q})] \cdot R - n)$;
    $\widehat{K}[l] \leftarrow \widehat{K}[l] + \frac{L'}{nL} \cdot \widehat{\mathcal{S}}_{\mathcal{D}}[i, h_i(\boldsymbol{q})]$;

**return** $\widehat{\text{KDE}}_{\mathcal{D}}(\boldsymbol{q}) \leftarrow \text{Median}(\widehat{K}[1], \cdots, \widehat{K}[L'])$;

---

unbiased estimators for $\text{KDE}_{\mathcal{D}}(\boldsymbol{q})$ from $\mathcal{S}_{\mathcal{D}}$. Finally, these estimators are divided into $L'$ groups, each containing $L/L'$ estimators. For each group, we compute the mean value $\widehat{K}[l]$ for $l \in \{1, \cdots, L'\}$ and return the median value as an approximation $\widehat{\text{KDE}}_{\mathcal{D}}(\boldsymbol{q})$ for $\text{KDE}_{\mathcal{D}}(\boldsymbol{q})$.

### 3.2 THEORETICAL ANALYSIS

Next, we analyze the privacy guarantee, approximation bound, and complexity of MLDP-KDE. Note that all proofs are omitted from the main paper due to space limitations and are provided in Appendix B.

**Privacy Analysis.** We start by defining the distance $d_{\text{hash}} : [1, R]^L \times [1, R]^L \mapsto [0, L]$ for two sequences of hash values as the count of different positions. Obviously, $d_{\text{hash}}(\cdot, \cdot)$ is metric because it is nonnegative, symmetric, and satisfies the triangle inequality. The following lemma shows that the GRR mechanism on any sequence of hash values provides mLDP on $d_{\text{hash}}(\cdot, \cdot)$.

**Lemma 1.** *The GRR mechanism $\mathcal{M}_{\text{GRR}}$ with a privacy parameter $\gamma > 0$ provides $(\gamma d_{\text{hash}}, 0)$-mLDP on a sequence of $L$ integers in the range of $[1, R]$.*

For each $\boldsymbol{x} \in \mathcal{D}$, the LSH+GRR mechanism to produce $\widehat{H}(\boldsymbol{x})$ in Algorithm 1 is also $(\gamma d_{\text{hash}}, 0)$-mLDP because $H(\boldsymbol{x})$ must be an input for $\mathcal{M}_{\text{GRR}}$ in Lemma 1. Formally, for any $\boldsymbol{x}, \boldsymbol{x}' \in \mathbb{R}^m$ and $\boldsymbol{y} \in [1, R]^L$,

$$\mathcal{L}_{\boldsymbol{x}, \boldsymbol{x}'} = \ln \left( \frac{\Pr[\widehat{H}(\boldsymbol{x}) = \boldsymbol{y}]}{\Pr[\widehat{H}(\boldsymbol{x}') = \boldsymbol{y}]} \right) \leq \gamma d_{\text{hash}}\big( H(\boldsymbol{x}), H(\boldsymbol{x}') \big). \quad (3)$$

Then, we define a random variable $X$ for the distribution of $d_{\text{hash}}(H(\boldsymbol{x}), H(\boldsymbol{x}'))$ over all possible LSH functions in the 2-stable LSH scheme, and show that $X$ is binomial.

---

it applies the LSH computation [Coleman and Shrivastava, 2020] and the GRR mechanism [Kairouz et al., 2016] in a sequential manner.

When receiving the perturbed hash values from all users, the server builds the sketch $\mathcal{S}_{\mathcal{D}}$ similarly to that of the RACE sketch [Coleman and Shrivastava, 2020]. It initializes an array $L \times R$ of all zeros. For each sequence $\widehat{H}(\boldsymbol{x})$ of the hash values for $\boldsymbol{x} \in \mathcal{D}$, it increments the counter $\mathcal{S}_{\mathcal{D}}[i, \widehat{h}_i(\boldsymbol{x})]$ for each $i \in \{1, \cdots, L\}$. After processing all data points in $\mathcal{D}$, it returns the sketch $\mathcal{S}_{\mathcal{D}}$.

**KDE Query Processing.** Algorithm 2 presents how the server processes a KDE query using the sketch $\mathcal{S}_{\mathcal{D}}$. Upon receiving a query point $\boldsymbol{q} \in \mathbb{R}^m$, the server first calculates a sequence of hash values $h_1(\boldsymbol{q}), \cdots, h_L(\boldsymbol{q})$ employing an identical sequence of $L$ hash functions and rehashing scheme outlined in Algorithm 1. Then, we provide an approximation $\widehat{\text{KDE}}_{\mathcal{D}}(\boldsymbol{q})$ of $\text{KDE}_{\mathcal{D}}(\boldsymbol{q})$ through the $L$ corresponding counters $\mathcal{S}_{\mathcal{D}}[1, h_1(\boldsymbol{q})], \cdots, \mathcal{S}_{\mathcal{D}}[L, h_L(\boldsymbol{q})]$. Subsequently, we analyze how the output distribution is affected by the rehashing scheme and the GRR mechanism to derive an unbiased estimator of $\frac{\mathcal{S}_{\mathcal{D}}[i, h_i(\boldsymbol{q})]}{n}$ from $\mathcal{S}_{\mathcal{D}}[i, h_i(\boldsymbol{q})]$ for each $i \in \{1, \cdots, L\}$ (see Lemma 3 in Section 3.2 for how the estimator is attained). This process produces $L$

**Lemma 2.** *Define a random variable $X = d_{\text{hash}}(H(\boldsymbol{x}),$ $H(\boldsymbol{x}'))$, where the $L$ LSH functions are drawn independently from the 2-stable LSH scheme. Then, $X$ follows a binomial distribution $\mathcal{B}(L, \frac{R-1}{R} \cdot (1 - k(\boldsymbol{x}, \boldsymbol{x}')))$.*

According to Eq. 3 and Lemma 2, the LSH+GRR mechanism is shown to provide mLDP by applying the Chernoff bound [Chernoff, 1952].

**Theorem 1.** *The LSH+GRR mechanism in Algorithm 1 provides $(d_\chi, \eta)$-mLDP, where $d_\chi(\boldsymbol{x}, \boldsymbol{x}') = \frac{\gamma c L(R-1)}{\omega R} \cdot$ $d(\boldsymbol{x}, \boldsymbol{x}') + \gamma\sqrt{\frac{L \ln(1/\eta)}{2}}$ for any $c \geq 0.8$ and $\eta \in (0, 1)$.*

Based on Theorem 1, we indicate how to decide the value of $\gamma$ in Algorithm 1 w.r.t. a privacy budget $\varepsilon > 0$. Since the level of privacy in mLDP varies with $d(\boldsymbol{x}, \boldsymbol{x}')$, we should calibrate it with a radius $r > 0$. That is, we require the privacy level to be at most $\varepsilon$ for any $d(\boldsymbol{x}, \boldsymbol{x}') \leq r$. As such, it guarantees that a point $\boldsymbol{x}$ is $\varepsilon$-indistinguishable from any point $\boldsymbol{x}'$ within a ball of radius $r$ centered at $\boldsymbol{x}$. To achieve this, we need to ensure that $d_\chi(\boldsymbol{x}, \boldsymbol{x}') \leq \varepsilon$ when $d(\boldsymbol{x}, \boldsymbol{x}') \leq r$. According to Theorem 1, the value of $\gamma$ in Algorithm 1 should be

$$\gamma \leq \varepsilon / \left(\frac{0.8rL(R-1)}{\omega R} + \sqrt{\frac{L \ln(1/\eta)}{2}}\right). \quad (4)$$

By applying the Chernoff bound with the Kullback–Leibler (KL) divergence, we obtain another mLDP guarantee for the LSH+GRR mechanism.

**Corollary 1.** *Let $p = \frac{R-1}{R} \cdot (1 - k(\boldsymbol{x}, \boldsymbol{x}'))$. For any $0 < s < 1 - p$, the LSH+GRR mechanism provides $(d_\chi, \eta)$-mLDP, where $d_\chi(\boldsymbol{x}, \boldsymbol{x}') = \gamma L\left(\frac{c(R-1)}{\omega R} \cdot d(\boldsymbol{x}, \boldsymbol{x}') + s\right)$, $\eta = \exp\left(-L \cdot D_{\text{KL}}(p + s \parallel p)\right)$, and $c \geq 0.8$.*

Fixing $d_\chi(\boldsymbol{x}, \boldsymbol{x}') = r$, we can solve the two equations for $d_\chi(\boldsymbol{x}, \boldsymbol{x}')$ and $\eta$ in Corollary 1 using Newton's method to approximate the values of $s$ and $\gamma$. In practice, we compute the two $\gamma$'s according to Eq. 4 and Corollary 1 and use the larger one in Algorithm 1.

Finally, we present a worst-case privacy guarantee of the LSH+GRR mechanism that does not depend on the randomness of the 2-stable LSH scheme.

**Corollary 2.** *The LSH+GRR mechanism provides $\gamma L$-LDP.*

**Approximation Analysis.** According to [Coleman and Shrivastava, 2020], each initial counter provided by the sketch is an unbiased estimator for LSH kernels. However, this unbiasedness is no longer retained after rehashing and performing the GRR mechanism since the data distribution is changed. To provide an unbiased KDE, we need to analyze how the rehashing scheme and the GRR mechanism affect the collision probability of a query point $\boldsymbol{q}$ and any data point $\boldsymbol{x}$, as well as the distribution of each counter, and try

to recover the original estimator, as outlined in Algorithm 2. Next, we show that the estimator is unbiased and provides an upper bound of its variance.

**Lemma 3.** *For the estimator $\widehat{\mathcal{S}}_\mathcal{D}[i, h_i(\boldsymbol{q})]$ in Algorithm 2, it holds that $\mathbb{E}\left[\widehat{\mathcal{S}}_\mathcal{D}[i, h_i(\boldsymbol{q})]\right] = n\text{KDE}_\mathcal{D}(\boldsymbol{q})$ and*

$$\text{Var}\left[\widehat{\mathcal{S}}_\mathcal{D}[i, h_i(\boldsymbol{q})]\right] \leq \left(\frac{e^\gamma + R - 1}{e^\gamma - 1}\right)^2 \left(\frac{R}{R-1}\right)^2$$
$$\left(\sqrt{\frac{e^\gamma}{e^\gamma + R - 1} - \frac{1}{R}}\widetilde{K}(\boldsymbol{q}) + \frac{1}{\sqrt{R}}\right)^2, \quad (5)$$

*where $\widetilde{K}(\boldsymbol{q}) = \sum_{\boldsymbol{x} \in \mathcal{D}} \sqrt{k(\boldsymbol{x}, \boldsymbol{q})}$.*

By applying Chebyshev's inequality and the Chernoff bound to the output $\widehat{\text{KDE}}_\mathcal{D}(\boldsymbol{q})$ of Algorithm 2, which uses a common median-of-means technique for estimation, we obtain the following theorem for its approximation bound.

**Theorem 2.** *For the sketch $\mathcal{S}_\mathcal{D}$ constructed by Algorithm 1 with $L = O\left(\left(\frac{e^\gamma + R - 1}{e^\gamma - 1}\right)^2 \cdot \frac{\log(1/\eta)}{\alpha^2}\right)$ independent rows, the output $\widehat{\text{KDE}}_\mathcal{D}(\boldsymbol{q})$ of Algorithm 2 is guaranteed to be an $(\alpha, \eta)$-approximation of $\text{KDE}_\mathcal{D}(\boldsymbol{q})$.*

We note that the restrictions on the values of $\gamma$ and $L$ in Eq. 4 for the privacy guarantee and in Theorem 2 for the approximation bound may not be satisfiable at the same time when the privacy parameter $\varepsilon$ is too small. This is because Eq. 4 restricts the upper bound of $\gamma$, but, in the meantime, Theorem 2 limits its lower bound. Consequently, the required ranges of $\gamma$ by Eq. 4 and Theorem 2 may not overlap each other. To eliminate the circular dependence on $\gamma$ and $L$ and thus reconcile Eq. 4 and Theorem 2, we further establish the following approximation bound.

**Theorem 3.** *For the privacy parameter $\varepsilon = O\left(\frac{\log(1/\eta)}{\alpha^2}\right)$ and the sketch parameters $L = O\left(\frac{\log(1/\eta)}{\alpha^2}\right)$ and $R = O(1)$, the output $\widehat{\text{KDE}}_\mathcal{D}(\boldsymbol{q})$ of Algorithm 2 is guaranteed to be an $(\alpha, \eta)$-approximation of $\text{KDE}_\mathcal{D}(\boldsymbol{q})$.*

Theorem 3 implies that the approximation bound of MLDP-KDE might not hold when $\varepsilon = o\left(\frac{\log(1/\eta)}{\alpha^2}\right)$. In practice, we adopt a privacy-first strategy that determines the values of $\gamma$ and $L$ based on Eq. 4 or Corollary 1 to ensure the satisfaction of mLDP, albeit this may result in a smaller $L$ than required by the approximation bound in Theorem 2. This strategy achieves reasonable empirical performance, as the practical number of rows needed to estimate KDEs with small errors is much lower than the theoretical upper bound due to the conservatism of probability inequalities.

**Complexity Analysis.** In Algorithm 1, the server generates LSH parameters in $O(mL)$ time. Each user then computes and perturbs the hash values in $O(mL)$ time, followed by the server aggregating these sequences to build $\mathcal{S}_\mathcal{D}$ in $O(nL)$ time. Therefore, the server and each user take $O\left((m+n)L\right)$

(or simply $O(nL)$ as $n \gg m$) and $O(mL)$ time to build $\mathcal{S}_{\mathcal{D}}$, respectively. The total communication cost is $O(mnL)$. The spaces used to run Algorithm 1 are $O(nL)$ and $O(mL)$, and the size of $\mathcal{S}_{\mathcal{D}}$ is $O(LR)$.

On receiving a query $\boldsymbol{q}$, Algorithm 2 spends $O(mL)$ time to compute $\widehat{\mathrm{KDE}}_{\mathcal{D}}(\boldsymbol{q})$. The sketch size and query time in the MLDP-KDE framework are both sublinear w.r.t. $n$ because $L$ and $R$ are independent of $n$. For comparison, a non-sketch-based KDE method with LDP or mLDP takes shorter $O(n)$ and $O(m)$ pre-processing times on the server and user sides and has a lower communication cost of $O(nm)$ in the local computation model. However, the time and space complexities of processing each query without the sketch both increase significantly to $O(nm)$.

# 4  EXPERIMENTS

This section presents the empirical evaluation of MLDP-KDE on real-world and synthetic data sets.

## 4.1  EXPERIMENTAL SETUP

**Data Sets.** We employ the following four publicly available real-world data sets and one synthetic data set for performance evaluation.

- **CodRNA** [Uzilov et al., 2006] is a collection of RNA genomic sequences.

- **CovType** [Blackard, 1998] comprises different cartographic features of areas located in the Roosevelt National Forest.

- **RCV1** [Lewis et al., 2004] is an archive of categorized newswire stories from Reuters. We embed all the documents into a 100-dimensional Euclidean space.

- **Yelp**[3] includes reviews from users on Yelp. We represent users as 100-dimensional vectors, which are derived from a user-business rating matrix using NMF.

- **SYN**, created by `make_blobs` in the scikit-learn library,[4] comprises isotropic Gaussian blobs. We specify 10 centroids, each randomly drawn in the range $[-2, 2]^m$, and the standard deviation of each blob to $0.01$. We vary the number of data points ($n$ from $10^4$ to $10^6$) and the dimensions ($m$ from 5 to 50) to test scalability. By default, we set $n = 10^5$ and $m = 50$.

These data sets are commonly used to benchmark (non-private or private) KDE and clustering methods in the existing literature [Coleman and Shrivastava, 2021, Wagner et al., 2023]. We also note that they are not tailored to local privacy settings and that specialized data sets for KDE with

---

[3] https://www.yelp.com/dataset
[4] https://scikit-learn.org/

local privacy are still absent. Table 1 presents the statistics of these data sets where $\omega$ is the bandwidth of the $l_2$-LSH kernel, and $r$ is the calibration radius in the privacy calculation. According to [Coleman and Shrivastava, 2020, Wagner et al., 2023], we set $\omega$ based on the average distance $\bar{d}$ between two points in the data set, which is adjusted so that the average kernel density is around $0.1$. Following the common practice of mLDP [Fernandes et al., 2021], the value of $r$ is determined by computing the average distance $\tilde{d}$ from a point to its 100-th nearest neighbor and rounding it to two significant figures. We also provide additional experiments to evaluate how the value of $r$ affects the performance of MLDP-KDE in Appendix C.1.

Table 1: Statistics of data sets used in the experiments.

| Data Set | $n$ | $m$ | $\omega$ | $r$ |
|---|---|---|---|---|
| CodRNA | $488,565$ | 8 | $0.25$ | $0.1$ |
| CovType | $581,012$ | 55 | $0.5$ | $0.1$ |
| RCV1 | $804,414$ | 100 | $0.25$ | $0.2$ |
| Yelp | $1,986,079$ | 100 | $0.5$ | $0.0025$ |
| SYN | $10^4$–$10^6$ | 5–50 | $\sqrt{m}$ | $0.015 \cdot \sqrt{m}$ |

**Algorithms and Implementations.** We compare MLDP-KDE with the following five algorithms: RACE [Coleman and Shrivastava, 2020] is a non-private sketch method for the KDE problem; DM [Duchi et al., 2013], PM [Wang et al., 2019], and SW [Li et al., 2020] are LDP methods for numerical data publication and distribution estimation; GI [Andrés et al., 2013, Alvim et al., 2018] is a method to preserve location privacy in two-dimensional Euclidean space, which is extended to support higher-dimensional Euclidean distance by Fernandes et al. [2021]. Since SW only supports one-dimensional data, we extend it to multidimensional data by independently perturbing each dimension of a point using a privacy budget of $\frac{\varepsilon}{m}$. For DM, PM, SW, and GI, which are not customized for KDE, we employ the following adaptation: Each client perturbs their data points and sends them to the server; the server computes the KDE for a query point using these perturbed points. We refer to the above methods with this adapted procedure as **DM-KDE**, **PM-KDE**, **SW-KDE**, and **GI-KDE**, respectively. DP-KDE methods [Aldà and Rubinstein, 2017, Coleman and Shrivastava, 2021, Wagner et al., 2023] are not compared since they are limited to centralized settings.

All these algorithms were implemented in Python 3. All methods were conducted on a desktop with an Intel® Core™ i7-10700K CPU @3.0GHz and 32GB RAM. Each method was run on a single thread in each experiment. Our code and data are publicly available at https://github.com/yz2022/mldp-kde.

**Performance Measures.** For each data set, we randomly choose 100 points to form the query set $\mathcal{Q}$ and use the rest as the data set $\mathcal{D}$. We evaluate the KDE quality of each method by the mean squared error (MSE) across all queries

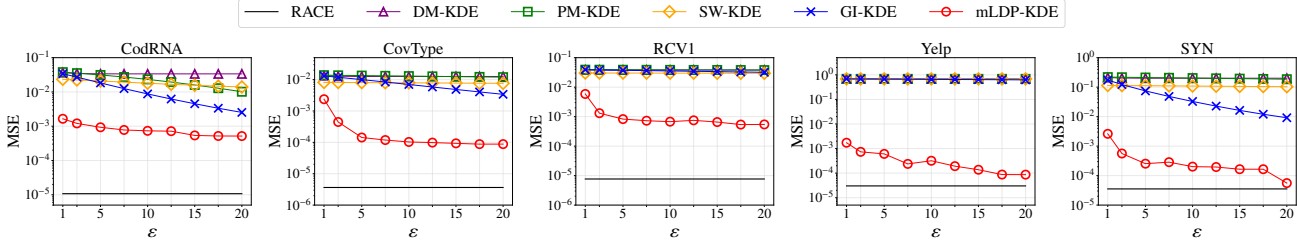

Figure 1: MSEs for KDE under LDP/mLDP with varying privacy budget $\varepsilon \in \{1, 2.5, 5, \cdots, 20\}$.

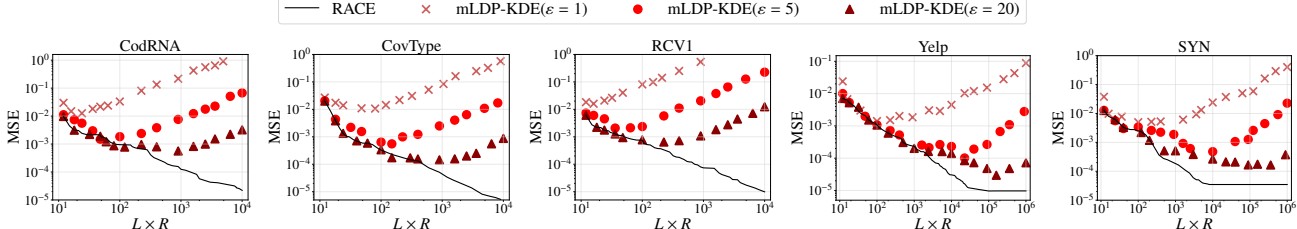

Figure 2: MSEs of RACE and MLDP-KDE for privacy budgets $\varepsilon = 1, 5, 20$ with varying sketch size $L \times R$.

in $\mathcal{Q}$, that is, $\text{MSE} = \frac{1}{|\mathcal{Q}|} \sum_{\boldsymbol{q} \in \mathcal{Q}} (\widehat{\text{KDE}}_{\mathcal{D}}(\boldsymbol{q}) - \text{KDE}_{\mathcal{D}}(\boldsymbol{q}))^2$. Given the stochastic nature of these methods, we run each experiment ten times with distinct (yet fixed) seeds and report the average result for each measure.

**Parameter Settings.** The values of various parameters were set as follows: (1) privacy budget $\varepsilon \in \{1, 2.5, 5, \cdots, 20\}$; (2) sketch height $L$ from 1 to $1,000$ and width $R$ from 2 to 100; (3) bandwidth $\omega$ and radius $r$ on each data set according to Table 1; (4) confidence parameter $\eta = 0.1$ and group parameter $L' = 1$. To decide the default values of $L$ and $R$, we run MLDP-KDE with different sketch sizes and use the ones with the lowest MSE for each privacy budget $\varepsilon$.

## 4.2 EXPERIMENTAL RESULTS

**Utility vs. Privacy.** We evaluate the performance of different algorithms in terms of the balance between the level of privacy and the quality of KDE. Figure 1 shows the MSE of the KDE query results returned by each algorithm, with the privacy budget $\varepsilon$ ranging from 1 to 20. For MLDP-KDE, we report the lowest MSE across different sketch sizes for each $\varepsilon$ value on every data set. For RACE, which does not involve data perturbation, we fix $L = 1,000$ and $R = 100$ and represent its result as a horizontal line in each plot. The difference between MLDP-KDE and RACE highlights the impact of the LSH+GRR mechanism on the quality of KDE.

In general, we observe that all LDP and mLDP algorithms exhibit a reduction in MSEs as the privacy budget $\varepsilon$ increases, indicating more accurate KDEs. MLDP-KDE significantly and consistently outperforms all baselines in terms of privacy-utility trade-offs. A key factor is that MLDP-KDE provides LDP for each point w.r.t. other points within

a distance of $r$, which better preserves the original data distribution and thus produces approximate KDE results of higher quality than the LDP baselines that should provide much more stringent LDP guarantee w.r.t. all possible points. The fact that MLDP-KDE significantly outperforms GI-KDE, which provides the same mLDP guarantee, indicates that adding noise to the hash values rather than the original data greatly reduces the privacy budget required to achieve the same level of utility. We find that the KDE quality of GI-KDE is highly dependent on the dimensionality $m$. This limitation comes from the exponential growth of the expected perturbation distance with increasing $m$, which makes the perturbed point further from the original point and thus spoils the data distribution. For higher-dimensional data sets such as RCV1 and Yelp, the MSEs of GI-KDE decrease more slowly, eventually resulting in estimates that are no better than those of the LDP methods.

**Utility vs. Sketch Size.** We test the effect of the sketch size $(L \times R)$ on the KDE quality of RACE and MLDP-KDE, sampling $1,000$ data points per data set for sketch construction and performing the same 100 KDE queries as in previous experiments. Figure 2 illustrates the MSEs of RACE and MLDP-KDE with privacy budgets $\varepsilon = 1, 5, 20$ with sketch sizes $L \times R$ from $10^1$ to $10^4$ (to $10^6$ on the Yelp and SYN data sets). As $L \times R$ increases, the MSE decreases significantly for both MLDP-KDE and RACE across all data sets. However, RACE and MLDP-KDE also show some differences: For RACE, the MSE finally stabilizes at a low level with increasing sketch sizes; but for MLDP-KDE, the MSE rebounds when the sketch size is too large, as the variances from GRR and the correction process outweigh the benefits of using more estimators and wider hash ranges. Furthermore, with small sketch sizes,

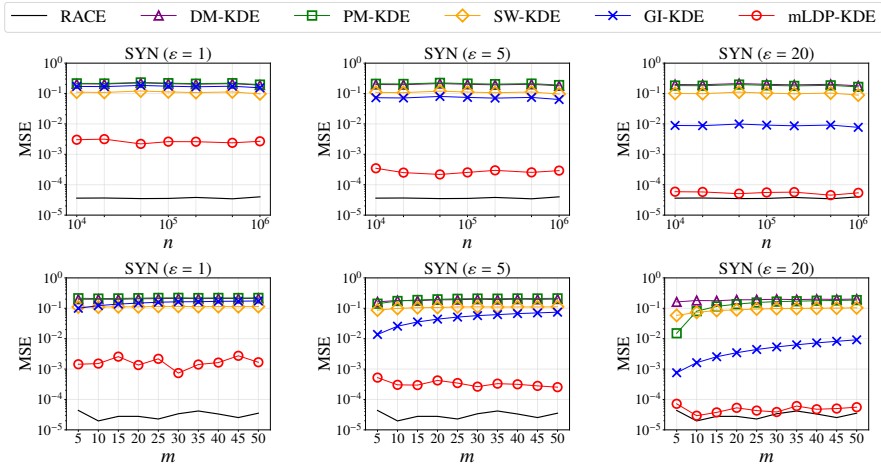

Figure 3: MSEs for KDE on the SYN data set with varying data set size $n$ from $10^4$ to $10^6$ and dimension $m$ from 5 to 50.

the MSEs of both algorithms are comparable due to the correction process during KDE query processing. Finally, for MLDP-KDE, using a larger sketch size leads to more benefits when the privacy budget is higher.

**Scalability Test.** We test the scalability of all methods on the SYN data sets, illustrating the MSE results for privacy budgets $\varepsilon = 1, 5, 20$ in Figure 3. The MSEs of all methods are stable regardless of $n$, indicating that the KDE quality is insensitive to the size of the data set. This confirms the results of Lemma 3 and Theorem 2, which indicate that the variance of MLDP-KDE is independent of $n$. With varying dimensionality $m$, MLDP-KDE exhibits slightly higher MSEs than RACE but outperforms other baselines in all cases and remains stable of different dimensions, indicating that MLDP-KDE can scale well to large-scale high-dimensional data sets. GI-KDE and PM-KDE can achieve a relatively low MSE for very small $m$ but cannot provide reasonable KDE results when $m > 5$.

Extended experimental results are omitted in the main paper due to space limits and will be provided in Appendix C.

## 5 CONCLUSION

In this paper, we address the challenge of answering KDE queries under LDP, where users privatize their data locally before sending them to an untrusted server. To ensure local privacy while preserving high-utility KDE results, we adopt a relaxed definition of LDP based on metrics called mLDP. Then, we propose a novel MLDP-KDE framework, which augments an LSH-based sketch method to provide unbiased estimations to KDE queries within a bounded additive error in sublinear time and space with high probability subject to mLDP. Experimental results on five data sets demonstrate the superiority of MLDP-KDE over existing KDE methods under both LDP and mLDP, especially showing significant advantages on large, high-dimensional data.

A current limitation of MLDP-KDE is that it tends to yield less promising KDE results in the high privacy regime, as indicated in Theorem 3 that MLDP-KDE might not achieve any approximation bound when $\varepsilon = o(\frac{\log(1/\eta)}{\alpha^2})$ and evidenced in Section 4 that MLDP-KDE provides KDEs with high errors when $\varepsilon \leq 1$. For future work, we intend to find better privacy and approximation bounds to improve its performance under tighter privacy budgets.

## Acknowledgements

We thank anonymous reviewers for their constructive comments to help improve this paper. This research was supported by the National Natural Science Foundation of China under Grant Numbers 62202169 and 62202170, the Open Research Fund of KLATASDS-MOE, ECNU, and the Ministry of Education, Singapore, under its MOE AcRF TIER 3 Grant (MOE-MOET32022-0001). Any opinions, findings, and conclusions or recommendations expressed in this material are those of the authors and do not reflect the views of the Ministry of Education, Singapore.

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

# A NOTATION TABLE

We summarize the symbols frequently used throughout the main paper in Table 2.

Table 2: List of frequently used symbols.

| Symbol | Description |
|---|---|
| $\mathcal{D}, \boldsymbol{x}$ | Data set in $\mathbb{R}^m$ and data point in $\mathcal{D}$ |
| $n, m$ | Size and dimensionality of $\mathcal{D}$ |
| $d(\cdot, \cdot)$ | (Metric) distance function |
| $\mathcal{H}, h(\cdot)$ | LSH scheme for $d(\cdot, \cdot)$ and LSH function drawn from $\mathcal{H}$ |
| $\omega$ | Bandwidth parameter in ($l_2$-LSH function) $h(\cdot)$ |
| $k(\cdot, \cdot)$ | LSH kernel function defined on $\mathcal{H}$ |
| $\mathrm{KDE}_{\mathcal{D}}(\boldsymbol{q})$ | Kernel density of $\mathcal{D}$ at query point $\boldsymbol{q}$ |
| $\widehat{\mathrm{KDE}}_{\mathcal{D}}(\boldsymbol{q})$ | Approximation of $\mathrm{KDE}_{\mathcal{D}}(\boldsymbol{q})$ |
| $\varepsilon$ | Privacy budget |
| $d_\chi(\cdot, \cdot)$ | Metric distance function for mLDP derived from $d(\cdot, \cdot)$ |
| $\alpha$ | Error parameter in approximation bound |
| $\eta$ | Confidence parameter in approximation and privacy bounds |
| $\mathcal{S}_{\mathcal{D}}$ | Sketch built on $\mathcal{D}$ for KDE queries |
| $L, R$ | Number & range of LSH functions (height & width of $\mathcal{S}_{\mathcal{D}}$) |
| $H, \widehat{H}$ | Sequences of hash values before and after perturbation |

# B MISSING PROOFS

We provide the proofs of theorems and lemmas that are missing from the main paper.

## B.1 PROOF OF LEMMA 1

*Proof.* Let $H = [h_1, \cdots, h_L]$, $H' = [h'_1, \cdots, h'_L] \in [1, R]^L$ be two length-$L$ integer sequences and $\boldsymbol{y} = [y_1, \cdots, y_L] \in [1, R]^L$ be any possible output sequence of $\mathcal{M}_{\mathrm{GRR}}$. We define a function $\phi(h, h')$ to indicate whether two integers in the same position of $H$ and $H'$ are equal, that is,

$$\phi(h, h') = \begin{cases} 1 & \text{if } h \neq h', \\ 0 & \text{otherwise.} \end{cases}$$

Based on the procedure of the GRR mechanism, we have

$$\Pr[\mathcal{M}_{\mathrm{GRR}}(H) = \boldsymbol{y}] = \prod_{i=1}^{L} p^{1-\phi(y_i, h_i)} q^{\phi(y_i, h_i)},$$

$$\Pr[\mathcal{M}_{\mathrm{GRR}}(H') = \boldsymbol{y}] = \prod_{i=1}^{L} p^{1-\phi(y_i, h'_i)} q^{\phi(y_i, h'_i)},$$

where $p = \frac{e^\gamma}{e^\gamma + R - 1}$ and $q = \frac{1}{e^\gamma + R - 1}$.

Consequently, we have

$$\ln\left(\frac{\Pr[\mathcal{M}_{\mathrm{GRR}}(H) = \boldsymbol{y}]}{\Pr[\mathcal{M}_{\mathrm{GRR}}(H') = \boldsymbol{y}]}\right) = \ln\left(\prod_{i=1}^{L} (\tfrac{p}{q})^{\phi(y_i, h'_i) - \phi(y_i, h_i)}\right) \leq \ln\left(\prod_{i=1}^{L} e^{\gamma \cdot \phi(h_i, h'_i)}\right) = \ln(e^{\gamma d_{\mathrm{hash}}(H, H')}) = \gamma d_{\mathrm{hash}}(H, H')$$

because $\phi(y_i, h'_i) - \phi(y_i, h_i) \leq \phi(h_i, h'_i)$ for any $i \in \{1, \cdots, L\}$ and $d_{\mathrm{hash}}(H, H') = \sum_{i=1}^{L} \phi(h_i, h'_i)$. Therefore, we prove that $\mathcal{M}_{\mathrm{GRR}}$ on any integer sequence of length $L$ and range $R$ provides $(\gamma d_{\mathrm{hash}}, 0)$-mLDP. $\square$

## B.2 PROOF OF LEMMA 2

*Proof.* According to [Datar et al., 2004], the collision probability of $\boldsymbol{x}$ and $\boldsymbol{x}'$ under the 2-stable LSH scheme is $k(\boldsymbol{x}, \boldsymbol{x}')$ in Eq. 1. After rehashing the original hash values randomly in the range $[1, R]$, the probability that two non-colliding hash values collide is $\frac{1}{R}$. Meanwhile, any colliding hash values must still collide after rehashing. Consequently, the collision probability of $h(\boldsymbol{x})$ and $h(\boldsymbol{x}')$ after rehashing is $k(\boldsymbol{x}, \boldsymbol{x}') + \frac{1}{R}(1 - k(\boldsymbol{x}, \boldsymbol{x}'))$. For the hash values $h(\boldsymbol{x})$ and $h(\boldsymbol{x}')$ after rehashing, we have $\Pr_{h \sim \mathcal{H}}[h(\boldsymbol{x}) \neq h(\boldsymbol{x}')] = \frac{R-1}{R} \cdot (1 - k(\boldsymbol{x}, \boldsymbol{x}'))$. That is, a random variable that indicates if $h(\boldsymbol{x}) \neq h(\boldsymbol{x}')$ follows a Bernoulli distribution with success probability $\frac{R-1}{R} \cdot (1 - k(\boldsymbol{x}, \boldsymbol{x}'))$. Based on the definition of $d_{\text{hash}}(\cdot, \cdot)$ and the fact that the LSH functions are independent of each other, $X$ follows a binomial distribution $\mathcal{B}\left(L, \frac{R-1}{R} \cdot (1 - k(\boldsymbol{x}, \boldsymbol{x}'))\right)$. $\square$

## B.3 PROOF OF THEOREM 1

> **Proposition 1** (Chernoff Bound [Chernoff, 1952]). *Let $X$ be a random variable drawn from a binomial distribution $\mathcal{B}(L, p)$. Then, for all $0 < s < 1 - p$, we have $\Pr[X \geq L(p + s)] \leq \exp\left(- L \cdot D_{\text{KL}}(p + s \parallel p)\right) \leq \exp(-2Ls^2)$, where $D_{\text{KL}}(p + s \parallel p) = (p + s) \ln \frac{p+s}{p} + (1 - p - s) \ln \frac{1-p-s}{1-p}$.*

*Proof.* Since $X \sim \mathcal{B}\left(L, \frac{R-1}{R} \cdot (1 - k(\boldsymbol{x}, \boldsymbol{x}'))\right)$ by Lemma 2, we have the following inequality from Proposition 1:

$$\Pr\left[X \geq L\left(\frac{R-1}{R} \cdot (1 - k(\boldsymbol{x}, \boldsymbol{x}')) + s\right)\right] \leq \exp(-2Ls^2). \tag{6}$$

By setting $s = \sqrt{\frac{\ln(1/\eta)}{2L}}$ in Eq. 6 for any $\eta \in (0, 1)$, we have $\Pr[\gamma X \geq \gamma L(\frac{R-1}{R} \cdot (1 - k(\boldsymbol{x}, \boldsymbol{x}')) + \sqrt{\frac{\ln(1/\eta)}{2L}})] \leq \eta$. Since $1 - k(\boldsymbol{x}, \boldsymbol{x}') \leq \frac{cd(\boldsymbol{x}, \boldsymbol{x}')}{\omega}$ for any $c \geq 0.8$, we further have

$$\Pr\left[\gamma X \geq \gamma L\left(\frac{c(R-1)d(\boldsymbol{x}, \boldsymbol{x}')}{\omega R} + \sqrt{\frac{\ln(1/\eta)}{2L}}\right)\right] \leq \eta. \tag{7}$$

Since $\mathcal{L}_{\boldsymbol{x}, \boldsymbol{x}'} \leq \gamma X$ as shown in Eq. 3, we obtain that

$$\Pr\left[\mathcal{L}_{\boldsymbol{x}, \boldsymbol{x}'} \geq \frac{\gamma c L(R-1)d(\boldsymbol{x}, \boldsymbol{x}')}{\omega R} + \gamma\sqrt{\frac{L \ln(1/\eta)}{2}}\right] \leq \eta. \tag{8}$$

Based on Definition 2, we prove that the LSH+GRR mechanism provides $(d_\chi, \eta)$-mLDP, where $d_\chi(\boldsymbol{x}, \boldsymbol{x}') = \frac{\gamma c L(R-1)}{\omega R} \cdot d(\boldsymbol{x}, \boldsymbol{x}') + \gamma\sqrt{\frac{L \ln(1/\eta)}{2}}$, for any $c \geq 0.8$ and $\eta \in (0, 1)$. $\square$

## B.4 PROOF OF COROLLARY 1

*Proof.* When applying the Chernoff bound with the Kullback–Leibler divergence in Proposition 1 to $X$, we have

$$\Pr[X \geq L(p + s)] \leq \exp\left(- L \cdot D_{\text{KL}}(p + s \parallel p)\right). \tag{9}$$

By applying the same procedure as for the proof of Theorem 1 on Eq. 9, we obtain

$$\Pr[\mathcal{L}_{\boldsymbol{x}, \boldsymbol{x}'} \geq \gamma L\left(\frac{c(R-1)d(\boldsymbol{x}, \boldsymbol{x}')}{\omega R} + s\right)] \leq \exp(-L \cdot D_{\text{KL}}(p + s \parallel p))$$

and conclude the proof. $\square$

## B.5 PROOF OF COROLLARY 2

*Proof.* For any $\boldsymbol{x}, \boldsymbol{x}' \in \mathbb{R}^d$, we have $d_{\text{hash}}(H(\boldsymbol{x}), H(\boldsymbol{x}')) \leq L$. According to [Chatzikokolakis et al., 2013], if a mechanism provides $(\gamma d_{\text{hash}}, 0)$-mLDP, then it will also provide $\gamma \max_{H, H'} d_{\text{hash}}(H, H')$-LDP. Therefore, the LSH+GRR mechanism satisfies $\gamma L$-LDP, which equals the total privacy budget of using the GRR mechanism with a privacy parameter $\gamma$ sequentially $L$ times. $\square$

## B.6 PROOF OF LEMMA 3

*Proof.* Due to the relationships between the 2-stable LSH scheme and the $l_2$-LSH kernel, the original collision probability $p_0(\boldsymbol{x}, \boldsymbol{q})$ of $\boldsymbol{x}$ and $\boldsymbol{q}$ before rehashing is exactly $k(\boldsymbol{x}, \boldsymbol{q})$. As already analyzed in the proof of Lemma 2, the collision probability $p_1(\boldsymbol{x}, \boldsymbol{q})$ of $\boldsymbol{x}$ and $\boldsymbol{q}$ after rehashing becomes

$$p_1(\boldsymbol{x}, \boldsymbol{q}) = k(\boldsymbol{x}, \boldsymbol{q}) + \tfrac{1}{R}(1 - k(\boldsymbol{x}, \boldsymbol{q})). \tag{10}$$

Next, if $\boldsymbol{x}$ and $\boldsymbol{q}$ collide, their collision probability after performing the GRR mechanism will be $\frac{e^\gamma}{e^\gamma + R - 1}$; otherwise, their probability of collision after performing the GRR mechanism will be $\frac{1}{e^\gamma + R - 1}$. Therefore, the collision probability $p_2(\boldsymbol{x}, \boldsymbol{q})$ of $\boldsymbol{x}$ and $\boldsymbol{q}$ after performing the GRR mechanism is

$$p_2(\boldsymbol{x}, \boldsymbol{q}) = \tfrac{p_1(\boldsymbol{x}, \boldsymbol{q}) \cdot e^\gamma}{e^\gamma + R - 1} + \tfrac{1 - p_1(\boldsymbol{x}, \boldsymbol{q})}{e^\gamma + R - 1}. \tag{11}$$

Based on Eqs. 10 and 11, we have

$$
\begin{aligned}
k(\boldsymbol{x}, \boldsymbol{q}) &= \big( \tfrac{(e^\gamma + R - 1) \cdot p_2(\boldsymbol{x}, \boldsymbol{q}) - 1}{e^\gamma - 1} - \tfrac{1}{R} \big) \cdot \tfrac{R}{R - 1} \\
&= \tfrac{(e^\gamma + R - 1)(R \cdot p_2(\boldsymbol{x}, \boldsymbol{q}) - 1)}{(e^\gamma - 1)(R - 1)}.
\end{aligned}
\tag{12}
$$

According to the sketch construction procedure, it is obvious that $\mathbb{E}[\mathcal{S}_\mathcal{D}[i, h_i(\boldsymbol{q})]] = \sum_{\boldsymbol{x} \in \mathcal{D}} p_2(\boldsymbol{x}, \boldsymbol{q})$. By replacing $p_2(\boldsymbol{x}, \boldsymbol{q})$ in Eq. 12 with $\mathcal{S}_\mathcal{D}[i, h_i(\boldsymbol{q})]$, summing up the results for all $\boldsymbol{x} \in \mathcal{D}$, and considering Algorithm 2, we have

$$\mathbb{E}\big[\widehat{\mathcal{S}}_\mathcal{D}[i, h_i(\boldsymbol{q})]\big] = \sum_{\boldsymbol{x} \in \mathcal{D}} k(\boldsymbol{x}, \boldsymbol{q}) = n\mathrm{KDE}_\mathcal{D}(\boldsymbol{q}). \tag{13}$$

According to Algorithm 2, we have

$$\mathrm{Var}\big[\widehat{\mathcal{S}}_\mathcal{D}[i, h_i(\boldsymbol{q})]\big] = \big( \tfrac{(e^\gamma + R - 1)R}{(e^\gamma - 1)(R - 1)} \big)^2 \cdot \mathrm{Var}\big[\mathcal{S}_\mathcal{D}[i, h_i(\boldsymbol{q})]\big]. \tag{14}$$

To compute $\mathrm{Var}\big[\mathcal{S}_\mathcal{D}[i, h_i(\boldsymbol{q})]\big]$, we define a random variable $I(\boldsymbol{x}, \boldsymbol{q})$ to indicate if $\widehat{h}_i(\boldsymbol{x}) = h_i(\boldsymbol{q})$, that is,

$$
I(\boldsymbol{x}, \boldsymbol{q}) = \begin{cases} 1 & \text{if } \widehat{h}_i(\boldsymbol{x}) = h_i(\boldsymbol{q}), \\ 0 & \text{otherwise.} \end{cases}
\tag{15}
$$

We can see that $I(\boldsymbol{x}, \boldsymbol{q})$ is a Bernoulli variable with success probability $p_2(\boldsymbol{x}, \boldsymbol{q})$ and $\mathcal{S}_\mathcal{D}[i, h_i(\boldsymbol{q})] = \sum_{\boldsymbol{x} \in \mathcal{D}} I(\boldsymbol{x}, \boldsymbol{q})$. Since $\mathrm{Var}[X] = \mathbb{E}[X^2] - \mathbb{E}[X]^2$ for any random variable $X$,

$$\mathrm{Var}\big[\mathcal{S}_\mathcal{D}[i, h_i(\boldsymbol{q})]\big] \le \mathbb{E}\big[(\sum_{\boldsymbol{x} \in \mathcal{D}} I(\boldsymbol{x}, \boldsymbol{q}))^2\big]. \tag{16}$$

Then, we acquire that

$$
\begin{aligned}
\mathbb{E}\big[(\sum_{\boldsymbol{x} \in \mathcal{D}} I(\boldsymbol{x}, \boldsymbol{q}))^2\big] &= \sum_{\boldsymbol{x} \in \mathcal{D}} \sum_{\boldsymbol{x}' \in \mathcal{D}} \mathbb{E}\big[I(\boldsymbol{x}, \boldsymbol{q}) I(\boldsymbol{x}', \boldsymbol{q})\big] \\
&\le \sum_{\boldsymbol{x} \in \mathcal{D}} \sum_{\boldsymbol{x}' \in \mathcal{D}} \sqrt{\mathbb{E}\big[I^2(\boldsymbol{x}, \boldsymbol{q})\big] \mathbb{E}\big[I^2(\boldsymbol{x}', \boldsymbol{q})\big]} \\
&= \sum_{\boldsymbol{x} \in \mathcal{D}} \sum_{\boldsymbol{x}' \in \mathcal{D}} \sqrt{\mathbb{E}\big[I(\boldsymbol{x}, \boldsymbol{q})\big] \mathbb{E}\big[I(\boldsymbol{x}', \boldsymbol{q})\big]} \\
&= \big( \sum_{\boldsymbol{x} \in \mathcal{D}} \sqrt{p_2(\boldsymbol{x}, \boldsymbol{q})} \big)^2,
\end{aligned}
\tag{17}
$$

where the inequality follows from the Cauchy-Schwarz inequality. By combining Eqs. 16 and 17, we have

$$\mathrm{Var}\big[\mathcal{S}_\mathcal{D}[i, h_i(\boldsymbol{q})]\big] \le \big( \sum_{\boldsymbol{x} \in \mathcal{D}} \sqrt{p_2(\boldsymbol{x}, \boldsymbol{q})} \big)^2. \tag{18}$$

According to Eq. 12, we have

$$p_2(\boldsymbol{x}, \boldsymbol{q}) = \big( \tfrac{e^\gamma}{e^\gamma + R - 1} - \tfrac{1}{R} \big) \cdot k(\boldsymbol{x}, \boldsymbol{q}) + \tfrac{1}{R}. \tag{19}$$

By taking Eq. 19 into Eq. 18 and letting $t_1 = \frac{e^\gamma}{e^\gamma + R - 1} - \frac{1}{R}$ and $t_2 = \frac{1}{R}$, we further obtain that

$$
\begin{aligned}
\mathrm{Var}\big[\mathcal{S}_\mathcal{D}[i, h_i(\boldsymbol{q})]\big] &\le \big( \sum_{\boldsymbol{x} \in \mathcal{D}} \sqrt{t_1 \cdot k(\boldsymbol{x}, \boldsymbol{q}) + t_2} \big)^2 \\
&\le \big( \sqrt{t_1} \cdot \widetilde{K}(\boldsymbol{q}) + \sqrt{t_2} \big)^2,
\end{aligned}
\tag{20}
$$

where $\widetilde{K}(\boldsymbol{q}) = \sum_{\boldsymbol{x} \in \mathcal{D}} \sqrt{k(\boldsymbol{x}, \boldsymbol{q})}$. By combining Eq. 20 with Eq. 14, we finally acquire Eq. 5 and conclude the proof. $\square$

## B.7 PROOF OF THEOREM 2

*Proof.* The median-of-mean technique has been widely used to estimate the expected value of a random variable $X$ within an additive error $\alpha > 0$ with a failure probability of at most $\eta \in (0, 1)$. By applying Chebyshev's inequality and the Chernoff bound, we find that when $L = O\left(\frac{\mathrm{Var}[X] \cdot \log(1/\eta)}{\alpha^2}\right)$ samples are drawn from the distribution of $X$, the median-of-mean estimator $\widehat{X}$ will satisfy that $\Pr\left[|\widehat{X} - \mathbb{E}[X]| \leq \alpha\right] \geq 1 - \eta$. Based on Lemma 3, we find that the expected value of each $\frac{1}{n} \cdot \widehat{\mathcal{S}}_{\mathcal{D}}[i, h_i(\boldsymbol{q})]$ is $\mathrm{KDE}_{\mathcal{D}}(\boldsymbol{q})$ and its variance is bounded. In addition, since $0 < \sqrt{\frac{e^\gamma}{e^\gamma + R - 1}} - \frac{1}{R} < 1$, $0 \leq \widetilde{K}(\boldsymbol{q}) \leq n$, and $\frac{1}{\sqrt{R}} < 1$ (for $R > 1$), the variance of $\frac{1}{n} \cdot \widehat{\mathcal{S}}_{\mathcal{D}}[i, h_i(\boldsymbol{q})]$ can be simplified as

$$\mathrm{Var}[\tfrac{1}{n} \cdot \widehat{\mathcal{S}}_{\mathcal{D}}[i, h_i(\boldsymbol{q})]] \leq \left(\tfrac{(e^\gamma + R - 1)R}{(e^\gamma - 1)(R - 1)}\right)^2 \cdot \left(\tfrac{n+1}{n}\right)^2.$$

Then, since $\left(\frac{R}{R-1}\right)^2 \leq 4$ and $\left(\frac{n+1}{n}\right)^2 \leq 4$, we have

$$\mathrm{Var}[\tfrac{1}{n} \cdot \widehat{\mathcal{S}}_{\mathcal{D}}[i, h_i(\boldsymbol{q})]] = O\left(\left(\tfrac{e^\gamma + R - 1}{e^\gamma - 1}\right)^2\right).$$

Therefore, we conclude that the median of means of $L = O\left(\left(\frac{e^\gamma + R - 1}{e^\gamma - 1}\right)^2 \cdot \frac{\log(1/\eta)}{\alpha^2}\right)$ independent estimators in the form of $\frac{1}{n} \cdot \widehat{\mathcal{S}}_{\mathcal{D}}[i, h_i(\boldsymbol{q})]$ is an $(\alpha, \eta)$-approximation of $\mathrm{KDE}_{\mathcal{D}}(\boldsymbol{q})$. □

## B.8 PROOF OF THEOREM 3

*Proof.* First, by simplifying Eq. 4, we get $\gamma = \frac{\varepsilon}{O(L + \sqrt{L \log(1/\eta)})}$, since $r$ and $\omega$ are data-dependent constants and $0.5 \leq \frac{R-1}{R} < 1$. Taking $\varepsilon = O\left(\frac{\log(1/\eta)}{\alpha^2}\right)$ and $L = O\left(\frac{\log(1/\eta)}{\alpha^2}\right)$ into the above equation, we have

$$\gamma = \frac{O\left(\frac{\log(1/\eta)}{\alpha^2}\right)}{O\left(\frac{\log(1/\eta)}{\alpha^2}\right) + O\left(\frac{\log(1/\eta)}{\alpha}\right)} = \frac{O\left(\frac{\log(1/\eta)}{\alpha^2}\right)}{O\left(\frac{\log(1/\eta)}{\alpha^2}\right)} = O(1) \tag{21}$$

Given that $\gamma = O(1)$ and $R = O(1)$, we can also simplify $L = O\left(\frac{\log(1/\eta)}{\alpha^2}\right)$ in Theorem 2. This means that Eq. 4 and Theorem 2 hold at the same time when $\varepsilon = O\left(\frac{\log(1/\eta)}{\alpha^2}\right)$, $L = O\left(\frac{\log(1/\eta)}{\alpha^2}\right)$, and $R = O(1)$ and the circular dependence on $\gamma$ and $L$ is resolved. □

# C ADDITIONAL EXPERIMENTAL RESULTS

## C.1 EFFECT OF PRIVACY RADIUS ON MLDP-KDE

We tested two schemes to decide the value of $r$ in the existing literature on mLDP: (1) setting $r$ to the maximum of the $x$-th percentile distance of a point from its neighbors for some $x \in (0, 100]$ [Chatzikokolakis et al., 2015] and (2) setting $r$ to the average distance from a point to its $t$-nearest neighbors for some $t \in \mathbb{Z}^+$ [Fernandes et al., 2021].

Initially, we set $r$ as the maximum of the 10th percentile distance between a point and its neighbors. Unfortunately, as presented in Table 3, such settings of $r$ resulted in sub-par KDE quality on most data sets, which is attributable to their highly skewed distributions, where a few outliers are distant from most other points and substantially increase the value of $r$. We then shifted our focus to adjust the value of $r$ based on the average, rather than the maximum, distance from a point to its $t$-nearest neighbors for $t = 1, 10, 100, 1000, 10000$. This yields more promising results, as detailed in Table 3. These findings suggest that MLDP-KDE can achieve high-quality KDE results while offering a reasonable level of privacy protection where each point is, on average, indistinguishable from $100$–$10,000$ other points in the data set. By default, we set the value of $r$ w.r.t. $t = 100$ on each data set in the remaining experiments.

## C.2 TIME EFFICIENCY

The first row of Figure 4 presents the construction time of each method on five data sets by varying the privacy budget from 1 to 20. We observe that RACE generally has the longest construction time. Compared to non-sketch methods, RACE

Table 3: MSEs of MLDP-KDE when the privacy radius $r$ is set to the average distance from a point to its $t$-nearest neighbors for $t \in \{1, 10, 100, 1000, 10000\}$ (rounded to two or three significant figures) or the maximum of the 10th percentile distance of a point from its neighbors.

| Data Set | $t$ | $r$ | MSE | | |
|---|---|---|---|---|---|
| | | | $\varepsilon = 1$ | $\varepsilon = 5$ | $\varepsilon = 20$ |
| CodRNA | 1 | 0.01 | 0.0021 | 0.0006 | 0.0003 |
| | 10 | 0.055 | 0.0018 | 0.0007 | 0.0003 |
| | 100 | 0.1 | 0.0016 | 0.0009 | 0.0005 |
| | 1,000 | 0.15 | 0.0044 | 0.00095 | 0.0006 |
| | 10,000 | 0.2 | 0.0044 | 0.00142 | 0.0006 |
| | (max of 10th percentile) | 0.8494 | 0.0268 | 0.00174 | 0.0009 |
| CovType | 1 | 0.01 | 0.0003 | 0.0002 | $6 \times 10^{-5}$ |
| | 10 | 0.06 | 0.0003 | 0.0001 | $8 \times 10^{-5}$ |
| | 100 | 0.1 | 0.0023 | 0.0001 | $8 \times 10^{-5}$ |
| | 1,000 | 0.3 | 0.0065 | 0.0004 | 0.0001 |
| | 10,000 | 0.5 | 0.0144 | 0.0005 | 0.0001 |
| | (max of 10th percentile) | 1.7803 | 0.0520 | 0.0085 | 0.0012 |
| RCV1 | 1 | 0.01 | 0.0008 | 0.0006 | 0.0002 |
| | 10 | 0.055 | 0.0013 | 0.0007 | 0.0002 |
| | 100 | 0.2 | 0.0058 | 0.0008 | 0.0006 |
| | 1,000 | 0.35 | 0.006 | 0.0008 | 0.0006 |
| | 10,000 | 0.5 | 0.022 | 0.0013 | 0.0007 |
| | (max of 10th percentile) | 0.8565 | 0.013 | 0.0013 | 0.0008 |
| Yelp | 1 | 0.001 | 0.0008 | 0.00034 | 0.00018 |
| | 10 | 0.00175 | 0.0014 | 0.00038 | $7 \times 10^{-5}$ |
| | 100 | 0.0025 | 0.0016 | 0.00056 | $7 \times 10^{-5}$ |
| | 1,000 | 0.00375 | 0.0013 | 0.00041 | $8 \times 10^{-5}$ |
| | 10,000 | 0.005 | 0.0015 | 0.00073 | 0.0001 |
| | (max of 10th percentile) | 8.4035 | 0.0295 | 0.0154 | 0.0101 |
| SYN | 1 | 0.072 | 0.0034 | 0.00027 | 0.0001 |
| | 10 | 0.088 | 0.0036 | 0.00035 | 0.0001 |
| | 100 | 0.107 | 0.0037 | 0.0008 | 0.0001 |
| | 1,000 | 0.142 | 0.0036 | 0.0008 | 0.0001 |
| | 10,000 | 0.177 | 0.0037 | 0.0004 | 0.0001 |
| | (max of 10th percentile) | 11.2314 | 0.0082 | 0.0038 | 0.0024 |

takes a longer time for $L = 1,000$ LSH computations than $m \leq 100$ perturbation operations per point. RACE also builds sketches much slower than MLDP-KDE in most cases because its sketch sizes are much larger than those of MLDP-KDE. MLDP-KDE exhibits only a longer construction time than other algorithms when the privacy budget is higher and the sketch size is larger. Due to the additional GRR procedure, MLDP-KDE becomes slower than RACE when $L > 400$ (resp. $L = 1,000$ for RACE). DM-KDE, PM-KDE, SW-KDE, and GI-KDE take less construction time than sketch-based methods, which are barely affected by $\varepsilon$ because their perturbation procedures are the same for all values of $\varepsilon$.

The second row of Figure 4 depicts the query time of each method for $\varepsilon = 1$ to 20. MLDP-KDE shows a substantial improvement in query efficiency compared to other algorithms. Its query time is more than four orders of magnitude faster than that of DM-KDE, PM-KDE, SW-KDE, and GI-KDE, which compute the KDE by evaluating the kernel functions for all perturbed points. Compared to RACE, MLDP-KDE exhibits a query time of almost an order of magnitude faster on all data sets except Yelp because it uses much smaller values of $L$ in the sketch and requires fewer LSH computations. On the Yelp data set, the query efficiency of MLDP-KDE is lower than RACE when $\varepsilon > 30$ since the values of $L$ in their sketches are close to each other but MLDP-KDE requires additional computations for correction.

In summary, although MLDP-KDE may not always have a notable advantage in construction time compared to other algorithms, its query time efficiency is exceptionally high, which aligns with the complexity analysis outlined in Section 3.2. Given that the sketch and other data structures are constructed just once, trading a longer construction time for a significantly lower query time is justifiable.

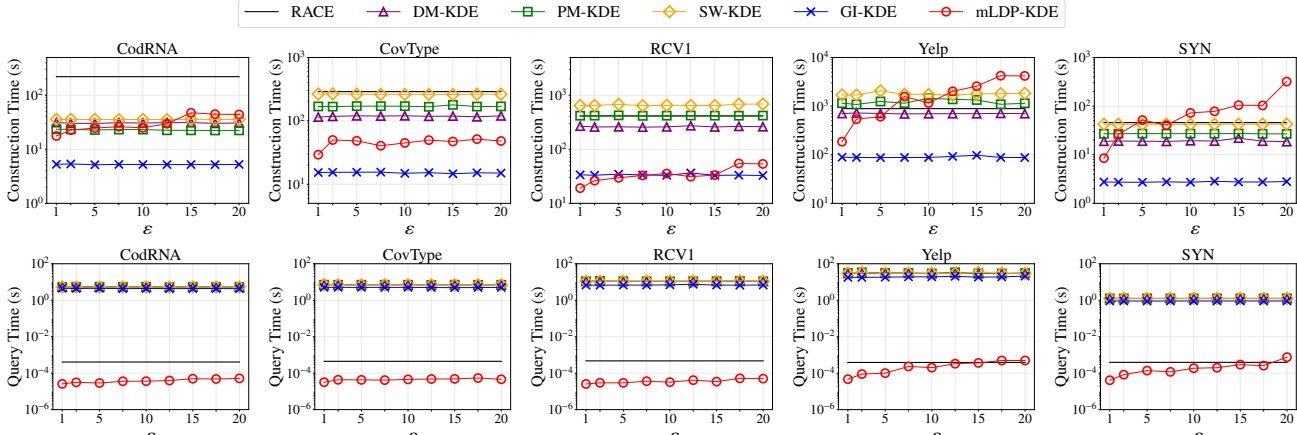

Figure 4: Construction time and query time of all algorithms for KDE under LDP/mLDP with varying privacy budget $\varepsilon \in \{1, 2.5, 5, \cdots, 20\}$.

## C.3 SKETCH SIZE AND COMMUNICATION COST

We show the sketch size and communication cost of MLDP-KDE with different privacy budgets on each data set in Figure 5. We find that both measures increase with the privacy budget $\varepsilon$. The size of the MLDP-KDE sketch increases naturally with $\varepsilon$ according to our privacy analysis in Section 3. For comparison, the RACE sketch size is always $8 \times 10^5$ bytes. The communication cost, which includes the transmission of the LSH parameters from the server to all clients and the hash sequences from all clients to the server, also increases with the privacy budget $\varepsilon$, since it is linear to the value of $L$. The communication costs of non-sketch methods are equal to the data set sizes listed in Table 1. As can be seen, sketch methods incur higher overhead in communication than non-sketch ones for $L > m$.

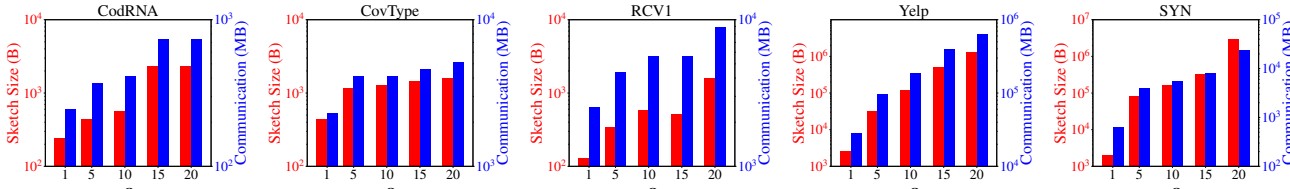

Figure 5: Sketch size and communication cost of MLDP-KDE with varying privacy budget $\varepsilon \in \{1, 5, 10, 15, 20\}$.

## C.4 ADDITIONAL SCALABILITY TEST

We test the scalability of different methods on the SYN data sets with numbers of points $n$ ranging from $10^4$ to $10^6$ (and $m = 50$) and dimensionalities $m$ ranging from 5 to 50 (and $n = 10^5$). Figure 6 shows the results for construction and query time. Since the construction time and query time of all methods except MLDP-KDE are not affected by $\varepsilon$, their results for $\varepsilon = 1, 5, 20$ are combined in Figure 6. The construction time of each method increases almost linearly with $n$. In terms of query time, MLDP-KDE and RACE are not affected by $n$, whereas non-sketch methods exhibit a linear increase with $n$. For different values of m, MLDP-KDE also outperforms all competitors in terms of construction and query time. Moreover, the time efficiency of MLDP-KDE depends mainly on the value of $\varepsilon$ but does not show obvious changes in different dimensions.

## C.5 VISUALIZED RESULTS

To verify that the KDE distributions generated by MLDP-KDE closely approximate the exact distributions, we performed some visualizations and showed the visualized results for the KDE distributions. We draw 2D heat maps of KDE distributions utilizing t-SNE for dimensionality reduction. These heat maps, presented in Figure 7, indicate that GI-KDE completely fails to preserve the exact KDE distributions in all cases, but MLDP-KDE generally preserves the exact KDE distributions in most cases when $\varepsilon = 5$ and always does when $\varepsilon = 20$. These results further substantiate the effectiveness of MLDP-KDE over GI-KDE.

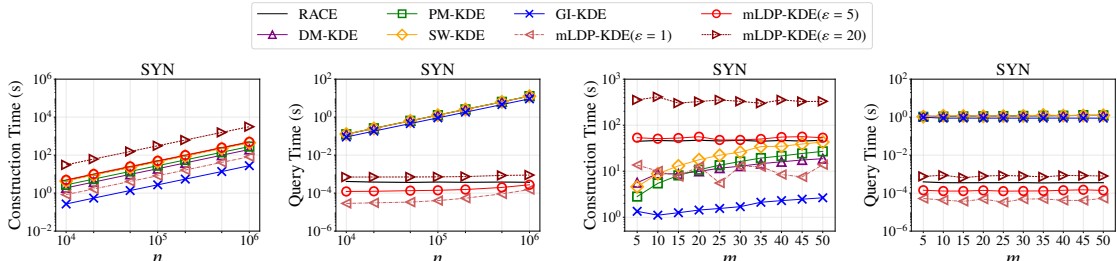

Figure 6: Construction and query time of all methods on the SYN data set with varying data set size $n$ from $10^4$ to $10^6$ and dimensionality $m$ from 5 to 50.

Figure 7: 2D heat maps for the KDE distributions provided by different methods on each data set, where t-SNE is used for dimensionality reduction.

# D  GENERALIZATIONS TO OTHER LSH KERNELS

We discuss how to extend MLDP-KDE to support other LSH kernels beyond the $l_2$-LSH kernel for the Euclidean distance.

$l_1$-**LSH Kernel.** By replacing the 2-stable LSH scheme with the 1-stable LSH scheme [Datar et al., 2004], where the random vector $\boldsymbol{a}$ in each LSH function is drawn from the Cauchy distribution instead of the Gaussian distribution, the MLDP-KDE framework can be applied directly to the $l_1$-LSH kernel for the Manhattan distance. The approximation bound and the complexity of MLDP-KDE are not affected after adaptation.

For the privacy analysis on the $l_1$-LSH kernel, denoted as

$$k_{l_1}(\boldsymbol{x}, \boldsymbol{x}') = \frac{2}{\pi} \arctan\left(\frac{\omega}{\|\boldsymbol{x} - \boldsymbol{x}'\|_1}\right) - \frac{\|\boldsymbol{x} - \boldsymbol{x}'\|_1}{\pi\omega} \ln\left(1 + \frac{\omega^2}{\|\boldsymbol{x} - \boldsymbol{x}'\|_1^2}\right),$$

where $\|\boldsymbol{x} - \boldsymbol{x}'\|_1$ is the $l_1$-distance of $\boldsymbol{x}$ and $\boldsymbol{x}'$, we have a slightly weaker bound $1 - k_{l_1}(\boldsymbol{x}, \boldsymbol{x}') \leq \frac{c_1}{\omega} \cdot \|\boldsymbol{x} - \boldsymbol{x}'\|_1 + c_2$ for any $c_1 \geq 1.2$ and $c_2 \geq 0.1$. Accordingly, the term $\frac{c(R-1)}{\omega R} \cdot d(\boldsymbol{x}, \boldsymbol{x}')$ in Theorem 1 and Corollary 1 should be replaced by $\frac{c_1(R-1)}{\omega R} \cdot d(\boldsymbol{x}, \boldsymbol{x}') + \frac{c_2(R-1)}{R}$ so that the LSH+GRR mechanism still provides mLDP.

**Angular Kernel.** To extend MLDP-KDE to the angular kernel [Coleman and Shrivastava, 2020, Lei et al., 2021], we replace the 2-stable LSH scheme with the SRP-LSH scheme. A function $h_{\mathrm{srp}} : \mathbb{R}^m \mapsto \{+1, -1\}$ in the SRP-LSH family is defined as $h_{\mathrm{srp}}(\boldsymbol{x}) = \mathrm{sign}(\boldsymbol{a} \cdot \boldsymbol{x})$, where $\boldsymbol{a}$ is also drawn from the standard $m$-dimensional Gaussian distribution. The angular kernel is defined as $k_{\mathrm{ang}}(\boldsymbol{x}, \boldsymbol{x}') = 1 - \frac{\theta(\boldsymbol{x}, \boldsymbol{x}')}{\pi}$, where $\theta(\boldsymbol{x}, \boldsymbol{x}')$ is the angle between $\boldsymbol{x}$ and $\boldsymbol{x}'$. We can see that, unlike $l_1$- and $l_2$-LSH functions, the output of each SRP-LSH function is binary. As such, by mapping the output values $\{-1, +1\}$ to the range $[1, 2]$, rehashing is not needed. Accordingly, the GRR mechanism is reduced to the special case of $R = 2$, i.e., the randomized response (RR) mechanism.

For privacy analysis, by removing rehashing and due to $k_{\mathrm{ang}}(\boldsymbol{x}, \boldsymbol{x}') = 1 - \frac{\theta(\boldsymbol{x}, \boldsymbol{x}')}{\pi}$, we refine Theorem 1 for the angular kernel as follows: The LSH+RR mechanism provides $(d_\chi, \eta)$-mLDP, where $d_\chi(\boldsymbol{x}, \boldsymbol{x}') = \frac{\gamma L}{\pi} \cdot \theta(\boldsymbol{x}, \boldsymbol{x}') + \gamma \sqrt{\frac{L \ln(1/\eta)}{2}}$. Corollary 1 holds by setting $p = \frac{\theta(\boldsymbol{x}, \boldsymbol{x}')}{\pi}$ and replacing the term $\frac{c(R-1)}{\omega R} \cdot d(\boldsymbol{x}, \boldsymbol{x}')$ with $\frac{\theta(\boldsymbol{x}, \boldsymbol{x}')}{\pi}$. Then, the unbiased estimator in Algorithm 2 for the angular kernel becomes

$$\widehat{\mathcal{S}}_{\mathcal{D}}[i, h_i(\boldsymbol{q})] = \frac{(e^\gamma + 1) \cdot \mathcal{S}_{\mathcal{D}}[i, h_i(\boldsymbol{q})] - 1}{e^\gamma - 1},$$

with its variance being bounded by

$$\left(\frac{e^\gamma + 1}{e^\gamma - 1}\right)^2 \cdot \left(\sqrt{\frac{e^\gamma - 1}{e^\gamma + 1}} \widetilde{K}(\boldsymbol{q}) + \sqrt{\frac{1}{e^\gamma + 1}}\right)^2.$$

Moreover, the number of independent rows required to guarantee an $(\alpha, \eta)$-approximate KDE in Theorem 2 remains $L = O\left((\frac{e^\gamma + 1}{e^\gamma - 1})^2 \cdot \frac{\log(1/\eta)}{\alpha^2}\right)$. Finally, the complexity of MLDP-KDE does not change after performing all the above adaptations.

**General LSH Kernel.** We finally analyze how MLDP-KDE can support KDE on general LSH kernels. Consider an LSH kernel defined by an LSH family $\mathcal{H}$ on a metric distance $d(\cdot, \cdot)$, where each $h \in \mathcal{H}$ maps a data point $\boldsymbol{x}$ to an integer $h(\boldsymbol{x})$. The range of $h(\cdot)$ can be bounded or unbounded, which is rehashed to the range of $[1, R]$. The basic property of $\mathcal{H}$ is that closer data points have a higher probability of mapping to the same hash value, which is formalized as follows.

**Definition 4** $((r_1, r_2, p_1, p_2)$-LSH family [Indyk and Motwani, 1998]). *A hash family $\mathcal{H}$ is $(r_1, r_2, p_1, p_2)$-LSH w.r.t. a metric distance $d(\cdot, \cdot)$, where $r_1 < r_2$ and $p_1 > p_2$, if for two points $\boldsymbol{x}$ and $\boldsymbol{x}'$:*

- *If $d(\boldsymbol{x}, \boldsymbol{x}') \leq r_1$, then $\mathrm{Pr}_{h \in \mathcal{H}}[h(\boldsymbol{x}) = h(\boldsymbol{x}')] \geq p_1$;*
- *If $d(\boldsymbol{x}, \boldsymbol{x}') \geq r_2$, then $\mathrm{Pr}_{h \in \mathcal{H}}[h(\boldsymbol{x}) = h(\boldsymbol{x}')] \leq p_2$.*

According to [Coleman and Shrivastava, 2020], the only additional requirement for any $(r_1, r_2, p_1, p_2)$-LSH family to form a positive semi-definite radial kernel in the form of $k(\boldsymbol{x}, \boldsymbol{x}') = f(d(\boldsymbol{x}, \boldsymbol{x}'))$ based on the collision probability function $f(\cdot)$ is that $f(\cdot)$ is monotonically decreasing. Intuitively, by applying any $(r_1, r_2, p_1, p_2)$-LSH family, MLDP-KDE can be used for the corresponding LSH kernel. Due to the definition of the LSH kernel, the approximation bounds of MLDP-KDE are naturally satisfied. For privacy analysis, unlike for a specific LSH kernel, since the relationship between $k(\boldsymbol{x}, \boldsymbol{x}')$ and $d(\boldsymbol{x}, \boldsymbol{x}')$ is unavailable, we can only provide probabilistic LDP bounds when $d(\boldsymbol{x}, \boldsymbol{x}')$ is in the ranges of $(0, r_1]$, $(r_1, r_2]$, and $(r_2, +\infty)$. Assuming that $\mathrm{Pr}_{\mathcal{H}}[h(\boldsymbol{x}) = h(\boldsymbol{x}')] = p_1$ when $d(\boldsymbol{x}, \boldsymbol{x}') = r_1$ and $\mathrm{Pr}_{\mathcal{H}}[h(\boldsymbol{x}) = h(\boldsymbol{x}')] = p_2$ when $d(\boldsymbol{x}, \boldsymbol{x}') = r_2$, we have

$$\mathrm{Pr}\left[\mathcal{L}_{\boldsymbol{x}, \boldsymbol{x}'} \geq \gamma L(\tfrac{R-1}{R}(1 - p_1) + s)\right] \leq \exp(-2Ls^2);$$
$$\mathrm{Pr}\left[\mathcal{L}_{\boldsymbol{x}, \boldsymbol{x}'} \geq \gamma L(\tfrac{R-1}{R}(1 - p_2) + s)\right] \leq \exp(-2Ls^2).$$

when $d(\boldsymbol{x}, \boldsymbol{x}') \leq r_1$ and $d(\boldsymbol{x}, \boldsymbol{x}') \leq r_2$. In this way, similar results to those of Theorem 1 and Corollary 1 can be obtained by replacing $s$ with the corresponding values. When $d(\boldsymbol{x}, \boldsymbol{x}') \geq r_2$, MLDP-KDE still provides $\gamma L$-LDP by Corollary 2.

## D.1 EXPERIMENTAL EVALUATION FOR OTHER LSH KERNELS

To demonstrate the generalizability of MLDP-KDE, we evaluate the MSEs of different methods for the $l_1$-LSH kernel and the angular kernel with privacy budgets $\varepsilon \in \{1, 2.5, 5, \cdots, 20\}$.

For the $l_1$-LSH kernel, we ran all methods except GI-KDE, which is specific to the Euclidean distance, on the CovType data set. The values of $\omega$ and $r$ are set to 2.5 and 0.05 for the $l_1$-LSH kernel because the average Manhattan distance between the points is about five times greater than the average Euclidean distance. As shown in the first plot of Figure 8, we can see that MLDP-KDE performs consistently and significantly better than other algorithms. However, its MSEs for the $l_1$-LSH kernel are generally higher because of larger constants in the privacy bound.

For the angular kernel, we carried out experiments on the Yelp data set and compared all methods with two additional baselines, FKM-LL-RACE and FKM-LR-RACE, which integrate the two LSH schemes under mLDP in [Fernandes et al., 2021] with RACE. From the second plot in Figure 8, we still observe that MLDP-KDE outperforms all other algorithms by a large margin. The advantage of MLDP-KDE over FKM-LL-RACE and FKM-LR-RACE further confirms its effectiveness for KDE problems compared to general LSH methods under mLDP.

These results support our justification for the generalization of MLDP-KDE to other LSH kernels.

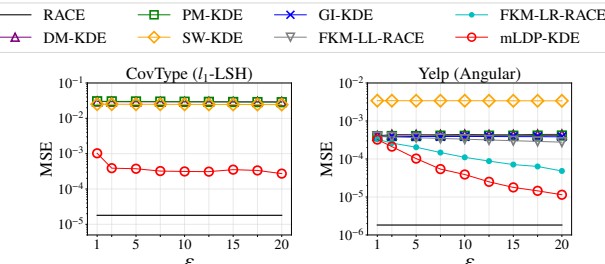

Figure 8: MSEs of different algorithms on the CovType data set for the $l_1$-LSH kernel and the Yelp data set for the angular kernel with varying privacy budgets $\varepsilon \in \{1, 2.5, 5, \cdots, 20\}$.