# OpenReview forum: "Approximate Kernel Density Estimation under Metric-based Local Differential Privacy"
_auai.org/UAI/2024/Conference — UAI 2024 poster_

### Official Review · Reviewer_CwuB · 2024-03-20

**Q2-1 Originality-Novelty:** 3
**Q2-2 Correctness-Technical Quality:** 3
**Q2-5 Clarity Of Writing:** 3

**Q1 Summary And Contributions:**

The paper shows how to do approximate, sublinear time KDE with metric differential privacy. Existing non-private methods use LSH with the dot product---the authors add a k-randomized response step to the LSH function to achieve metric DP. Accordingly, there is a privacy-dependent reduction in the accuracy guarantee of the algorithm which can be counteracted by using more hash functions. The authors then collect experimental data compared to several baselines, which demonstrates their results can outperform private benchmarks.

**Q2-3 Extent To Which Claims Are Supported By Evidence:**

3: Good: the main claims are supported by convincing evidence (in the form of adequate experimental evaluation, proofs, (pseudo-)code, references, assumptions).

**Q2-4 Reproducibility:**

3: Good: key resources (e.g. proofs, code, data) are available and key details (e.g. proofs, experimental setup) are sufficiently well-described for competent researchers to confidently reproduce the main results.

**Q3 Main Strengths:**

The behavior of the private LSH algorithm is rigorously bounded with a novel utility guarantee which accounts for the noise due to privacy. Also, the private LSH algorithm empirically outperforms the best-known metric DP algorithm for KDE.

**Q4 Main Weakness:**

The primary weakness of the paper is that there is not quite enough evidence to show that reasonable utility is possible with metric DP.  Theorem 2 says that utility gets better with higher L and worse with smaller \gamma, but by Equation 4, increasing L will reduce \gamma. Thus, it is currently hard to say if it is possible to set L to obtain a good privacy and utility guarantee, and I think the authors should investigate this question more. I have some hesitations about the experiments, too, since r is set to be the average distance to the \emph{nearest} neighbor, which means that privacy is only guaranteed for the nearest neighbor. I think this privacy guarantee is too weak. With such a small value of r, it is not surprising that the proposed mechanism outperforms the local DP baselines, leaving the most significant evidence, to me, being the improved utility against the metric DP baseline.

**Q5 Detailed Comments To The Authors:**

Definition 2 is a statement about probability, but there are no random variables in it. I believe you mean to say that y is a random variable drawn from M(x).

What is the difference between the two privacy guarantees in Theorem 1 and Corollary 1? Is it the case that one can sometimes be stronger than the other?

The utility guarantee in Theorem 2 is not explicit enough, yet. It says that if L is set higher than some function f(\gamma), the LSH will give good utility. However, equation 4 says that \gamma should be set smaller than g(L), which gives a circular dependence. To break this dependence, consider adding a statement to Theorem 3 that says "in particular, for a privacy budget $\epsilon$, and a particular choice L(\epsilon, \alpha, \eta), we obtain an (\alpha, \eta) approximate LSH scheme". This would then highlight the cost of privacy explicitly.

Might it be possible to strengthen the privacy guarantee of Theorem 1 with strong composition?

The question of how to set r for metric DP is tricky, but a starting point could be the maximum of the 10% percentile distance of a point to its neighbors, since this guarantees privacy for every point to at least 10% of its neighbors. The reference "Constructing elastic distinguishability metrics for location privacy" gives a more systematic way to do this for location data.

The experiments use \epsilon as high as 50, but it seems your mechanism offers good utility at \epsilon = 5. Consider increasing r to a more reasonable level, and then running the algorithms with smaller \epsilon (the baselines other GI-KDE don't perform well on this problem, so I don't think they even need to be plotted).

**Q9 Complying With Reviewing Instructions:**

Yes

---

> ### Author Rebuttal · Authors · 2024-04-09
>
> **(Rebuttal part 1 out of 2)**
>
> We are grateful to Reviewer CwuB for offering a range of insightful comments, which will significantly contribute to enhancing the technical quality of our work. Next, we will offer detailed explanations and address your concerns.
>
> **C1** Typos in Definition 2
>
> **Response:**
> Thank you for highlighting the oversight.
> Indeed, in Definition 2, it should be explicitly stated that $M(\cdot)$ is randomized. We will correct this typo in a revision opportunity.
>
> **C2** Difference between Theorem 1 and Corollary 1
>
> **Response:**
> Thank you for your query regarding the distinction between Theorem 1 and Corollary 1.
> The primary difference lies in their respective applications of the Chernoff bound: Theorem 1 utilizes the standard form, while Corollary 1 employs the KL-divergence form.
> We prioritize the result of Theorem 1 because it offers a closed-form expression and aligns more precisely with our mLDP definition.
> On the other hand, the result of Corollary 1 lacks a closed-form solution, necessitating the use of Newton's method for computing the value of $\gamma$.
> Notably, in practice, the result from Corollary 1 tends to yield a higher value of $\gamma$, offering a stronger practical bound.
> This is why we include Corollary 1 to augment Theorem 1, especially in determining the value of $\gamma$.
> We will elaborate on these nuances more thoroughly in any forthcoming revision.
>
> **C3** Circular dependence on $\gamma$ and $L$
>
> **Response:**
> Thank you for identifying a crucial aspect of our theoretical analysis.
> In response to your comments, we have endeavored to reconcile the results of Equation 4 and Theorem 2 to eliminate the circular dependence and establish a new bound (Theorem 3) for the values of $\varepsilon$, $L$, and $R$ in our method.
> Theorem 3 ensures an $(\alpha, \eta)$-approximation for KDE, which is detailed as follows:
>
> *Theorem 3: For a privacy parameter $\varepsilon = O(\frac{\log(1/\eta)}{\alpha^2})$ and sketch parameters $L = O(\frac{\log(1/\eta)}{\alpha^2})$ and $R = O(1)$, the output of Algorithm 2 is guaranteed to be an $(\alpha, \eta)$-approximation of $KDE_{D}(q)$.*
>
> First, by simplifying Eq. 4, we get $\gamma = \frac{\varepsilon}{O(L + \sqrt{L \log(1/\eta)})}$, since $r$ and $\omega$ are data-dependent constants and $0.5 \leq \frac{R - 1}{R} < 1$.
> Taking $\varepsilon = O(\frac{\log(1/\eta)}{\alpha^2})$ and $L = O(\frac{\log(1/\eta)}{\alpha^2})$ into the above equation, we have
> \begin{equation*}
>     \gamma = \frac{O(\frac{\log(1/\eta)}{\alpha^2})}{O(\frac{\log(1/\eta)}{\alpha^2} + \frac{\log(1/\eta)}{\alpha})} = \frac{O(\frac{\log(1/\eta)}{\alpha^2})}{O(\frac{\log(1/\eta)}{\alpha^2})} = O(1).
> \end{equation*}
> Given that $\gamma = O(1)$ and $R = O(1)$, $L$ in Theorem 2 can also be simplified to $L = O(\frac{\log(1/\eta)}{\alpha^2})$.
> This means that **Eq. 4 and Theorem 2 hold at the same time** when $\varepsilon = O(\frac{\log(1/\eta)}{\alpha^2})$, $L = O(\frac{\log(1/\eta)}{\alpha^2})$, and $R = O(1)$ and the circular dependence on $\gamma$ and $L$ is resolved.
>
> However, Theorem 3 indicates that the approximation bound of mLDP-KDE does not hold when $\varepsilon = o(\frac{\log(1/\eta)}{\alpha^2})$.
> In practice, we adopt a **privacy-first** strategy, determining the values of $\gamma$ and $L$ based on Eq. 4 or Corollary 1 to satisfy mLDP, although this may result in a smaller $L$ than required by the approximation bound in Theorem 2.
> Despite this, our strategy typically achieves reasonable empirical performance, as the practical number of rows needed for a KDE with small error is often lower than the theoretical upper bound due to the conservatism of probability inequalities.
> We will include these findings, their detailed proof, and corresponding discussions in the revised version of our paper.
>
> **C.4** Possibly better privacy guarantee
>
> **Response:**
> Thank you for your valuable suggestion regarding the enhancement of our theoretical analysis.
> While applying the advanced composition of differential privacy directly to Theorem 1 seems infeasible due to the intertwined nature of our LSH and GRR analyses, we do acknowledge the potential of utilizing advanced composition to derive better privacy bound in Corollary 2.
>
> Additionally, we have come across a very recent paper  [Liang and Yi, 2023] introducing a novel concept of geo-privacy based on concentrated differential privacy (CDP).
> We believe that the theoretical framework proposed by [Liang and Yi, 2023] might offer avenues to improve our method's privacy bound. However, we have not explored this possibility in depth due to time limits.
> We plan to present an enhanced privacy bound in Corollary 2 when a revision is allowed, leveraging the advanced composition.
> We also intend to mention [Liang and Yi, 2023] as a possible direction for future enhancement of our approach.
>
> **Additional References**
>
> Yuting Liang and Ke Yi. Concentrated geo-privacy. In CCS, page 1934–1948, 2023.

---

### Official Review · Reviewer_X2sy · 2024-03-22

**Q2-1 Originality-Novelty:** 2
**Q2-2 Correctness-Technical Quality:** 3
**Q2-5 Clarity Of Writing:** 3

**Q1 Summary And Contributions:**

This paper studies KDE under local constraints. It uses the mLDP definition of privacy and proposes mLDP-KDE algorithms.

**Q2-3 Extent To Which Claims Are Supported By Evidence:**

3: Good: the main claims are supported by convincing evidence (in the form of adequate experimental evaluation, proofs, (pseudo-)code, references, assumptions).

**Q2-4 Reproducibility:**

3: Good: key resources (e.g. proofs, code, data) are available and key details (e.g. proofs, experimental setup) are sufficiently well-described for competent researchers to confidently reproduce the main results.

**Q3 Main Strengths:**

1. The algorithms in Section 3 are well motivated.
2. Section 4 provides a theoretical analysis of the privacy guarantee and the utility of the framework. The proofs mostly rely on Chernoff-type bounds.

**Q4 Main Weakness:**

I have not identified any major weakness of this paper.

**Q5 Detailed Comments To The Authors:**

1. In the scalability test, the experiments suggest that "indicating that the KDE quality is insensitive to the size of the data set. " Can you provide more insights into why this happens?

2. The word LSH+GRR is in \emph{} the first time it appears and not italized when it appears later. Is there any reason for that? Also, the first time LSH+GRR appears, it is buried in the middle of Page 4 and not easy to find.

**Q9 Complying With Reviewing Instructions:**

Yes

---

> ### Author Rebuttal · Authors · 2024-04-09
>
> We sincerely thank Review X2sy for your insightful comments. Next, we will offer detailed explanations and address your concerns.
>
> **C.1** About the effect of data set size on KDE performance
>
> **Response:**
> We appreciate Reviewer X2sy for highlighting this important aspect.
> In typical non-private KDE or DP-KDE settings, as the dataset size $n$ increases, the variance of a KDE tends to decrease, typically by a factor of $O(\tfrac{1}{n})$, which usually results in an improvement in the quality of KDE.
> However, this does not apply in a local DP setting. Here, the perturbation applied to each data point injects a consistent level of variance into the KDE, making it independent of $n$ (as detailed in Lemma 3 and Theorem 2).
> Consequently, the quality of the KDE remains nearly unaffected by the number of data points used in the computation.
> We will elaborate further on this phenomenon in any future revision opportunity.
>
> **C.2** About inconsistent formats of LSH+GRR
>
> **Response:**
> We thank Reviewer X2sy for bringing attention to this presentation issue.
> We will standardize the LSH+GRR format to be non-italicized across the paper, except in instances within theorem-like environments where italicization is applied to the entire text.

---

### Official Review · Reviewer_pGyt · 2024-03-22

**Q2-1 Originality-Novelty:** 1
**Q2-2 Correctness-Technical Quality:** 1
**Q2-5 Clarity Of Writing:** 2

**Q1 Summary And Contributions:**

This paper evaluates Differential Privacy (DP) for kernel density estimation. The authors propose a locality-sensitive hashing-based sketch method that achieves ($\varepsilon$, $\delta$)-DP. The authors prove differential privacy bounds through first proving an intermediate result which can then imply local differential privacy.

**Q2-3 Extent To Which Claims Are Supported By Evidence:**

1: Poor: the authors fail to convincingly backup their main claims (e.g., if the experimental evaluation is flawed, proofs are lacking or invalid, references are missing, assumptions are not realistic, not specified, or not motivated).

**Q2-4 Reproducibility:**

2: Fair: key resources (e.g. proofs, code, data) are unavailable but key details (e.g. proof sketches, experimental setup) are sufficiently well-described for an expert to confidently reproduce the main results.

**Q3 Main Strengths:**

The authors are clear about what problem they are studying, the approaches they take to reach their conclusions, and reviewed a variety of relevant work.

**Q4 Main Weakness:**

The authors evaluate ($\varepsilon$, $\delta$)-DP in the case when $1\leq \varepsilon\leq 50$ in Figure 1 and Figure 2, suggesting the authors lack basic understanding of the definitions surrounding differential privacy problems and why people study ($\varepsilon$, $\delta$)-DP:

By definition of differential privacy, $1\leq \varepsilon\leq 50$ is putting a larger than 1 bound on a probability. I cannot be convinced that this is meaningful to look at.

**Q5 Detailed Comments To The Authors:**

I did not look further into the details of this paper after discovering the major flaw described in Main Weakness. If the authors can somehow convince me this is a misunderstanding, I am willing to dive further into the details provided in this paper.

**Q9 Complying With Reviewing Instructions:**

Yes

---

> ### Author Rebuttal · Authors · 2024-04-09
>
> **C.1** The authors evaluate $(\varepsilon, \delta)$-DP in the case when $1 \leq \varepsilon \leq 50$ in Figure 1 and Figure 2, suggesting the authors lack basic understanding of the definitions surrounding differential privacy problems and why people study $(\varepsilon, \delta)$-DP: By definition of differential privacy, $1 \leq \varepsilon \leq 50$ is putting a larger than $1$ bound on a probability. I cannot be convinced that this is meaningful to look at.
>
> **Response:**
> We respectfully disagree with this concern regarding our interpretation of $(\varepsilon, \delta)$-DP.
> In the standard definition, given by $\Pr[\mathcal{M}(x) \in S] \leq e^{\varepsilon} \cdot \Pr[\mathcal{M}(x') \in S] + \delta$, the parameter $\varepsilon$ is indeed required to be greater than $0$, with $\delta$ restricted to the range $[0, 1)$.
> However, our definition of mLDP is a variation of pure $\varepsilon$-DP, where $\delta = 0$ in the standard $(\varepsilon, \delta)$-DP, and $\varepsilon$ is defined in relation to a metric distance $d_{\chi}$ and is always positive.
> Furthermore, we incorporate the concept of probabilistic DP, which involves a confidence parameter $\eta$ in the range $(0, 1)$, as consistently reflected in our theoretical analyses and experiments.
> Therefore, our parameter setting of $1 \leq \varepsilon \leq 50$ is theoretically sound.
>
> It seems that there may be a misunderstanding with Reviewer pGyt, possibly conflating $\varepsilon$ with $\delta$.
> We acknowledge that our work involves several parameters in privacy and approximation analyses.
> We will further clarify each parameter's specific role and significance in our work.

---

### Official Review · Reviewer_Gde4 · 2024-03-24

**Q2-1 Originality-Novelty:** 3
**Q2-2 Correctness-Technical Quality:** 3
**Q2-5 Clarity Of Writing:** 4

**Q1 Summary And Contributions:**

The authors introduce the LDP-KDE framework, which augments a locality-sensitive hashing-based sketch method to provide mLDP while  answering any KDE query unbiasedly within an additive error with high probability in sublinear time and space. The paper is well written and well explained.

**Q2-3 Extent To Which Claims Are Supported By Evidence:**

3: Good: the main claims are supported by convincing evidence (in the form of adequate experimental evaluation, proofs, (pseudo-)code, references, assumptions).

**Q2-4 Reproducibility:**

4: Excellent: key resources (e.g. proofs, code, data) are available and key details (e.g. proof sketches, experimental setup) are comprehensively described for competent researchers to confidently and easily reproduce the main results.

**Q3 Main Strengths:**

Well written paper. The introduced methodology is clear and proofs are provided on key theorems.

**Q4 Main Weakness:**

Not many, however one of the key conclusions could be further explained (see Q5).

**Q5 Detailed Comments To The Authors:**

To aid reporducibility of the code more comments explaining the steps are referring to equations in the paper could be added.

If you were not in the centralised DP setting and your model was to be compared against DP-KDE differentiable privacy system - how would it compare on the datasets you have selected? Given this how would define your research question more clearly.

In your conclusion, also consider how this model could work under a global differentiable privacy framework.
With more data examples or theoretical proofs, can you explain this conclusion that you have come to:  A current limitation of MLDP-KDE is that it tends to yield less promising KDE results in the high privacy regime.

**Q9 Complying With Reviewing Instructions:**

Yes

---

> ### Author Rebuttal · Authors · 2024-04-09
>
> We sincerely thank Review Gde4 for appreciating our work. Next, we will offer detailed explanations and address your concerns.
>
> **C.1** About code clarity
>
> **Response:**
> We appreciate your recommendation to enhance the clarity of our code.
> During the rebuttal phase, we initiated two major enhancements: (1) we added essential comments to the key steps in our code, aligning them with our theoretical findings; (2) we refined the README file to clarify the relationships between scripts in our repository and the experiments described in our paper.
>
> **C.2** About selected datasets
>
> **Response:**
> Thank you for highlighting this important question.
> Our selection or generation of datasets is guided by two primary criteria: (1) availability/reproducibility and (2) frequent usage to benchmark the performance of KDE methods.
> This not only facilitates a comprehensive analysis of privacy-utility trade-offs but also aligns our research with established ones.
> Furthermore, we observed a lack of public datasets tailored to local DP settings.
>
> In common practice, whether for KDE, clustering, or other unsupervised learning scenarios such as [Coleman and Shrivastava, 2021;  Lei et al., 2021; Wagner et al., 2023] as well as [Chang et al., 2021], similar benchmark datasets are often used, regardless of non-private, centralized DP, and local DP methods.
> When there is a chance of revision, we plan to more explicitly outline our dataset selection rationale and underscore the notable absence of specialized local DP benchmark datasets.
>
> **C.3** Suggestions on Conclusion
>
> **Response:**
> Thank you for these insightful comments.
> Firstly, regarding the applicability of our model under a global, centralized DP framework, we confirm that it is compatible with the same metric-based relaxation.
> Second, our conclusion about the limited effectiveness of mLDP-KDE in high-privacy settings (``less promising KDE results in the high-privacy regime'') stems from experimental observations, particularly the increase in MSEs when $\varepsilon \leq 1$.
>
> In this rebuttal, we supplement our findings with additional visualized results (refer to the provided URLs for more details).
> Furthermore, this conclusion is cross-validated by our new theoretical analysis (in response to **C.3 of Reviewer CwuB**), indicating that the approximation and privacy guarantees of mLDP-KDE might not hold at the same time, particularly when $\varepsilon = o(\frac{\log(1/\eta)}{\alpha^2})$.
>
> **Additional References**
>
> Alisa Chang, Badih Ghazi, Ravi Kumar, and Pasin Manurangsi. Locally private k-means in one round. In Proceedings of
> the 38th International Conference on Machine Learning (ICML), pages 1441–1451, 2021.

---

### Meta-Review · Area_Chair_DS8r · 2024-04-15

Thank you for submitting your work to UAI. The problem formulation of approximate KDE under metric LDP and results are both within scope and novel. The authors have also addressed several suggestions from the reviewers, which should substantially strengthen the paper. These include running new experiments, updating the main theoretical result to be more explicit, and clarifying the writing and interpretation of results. I do not see remaining complaints about the substance of the paper.